# Spatial dynamics of malaria transmission

**Sean L. Wu**[1☯], **John M. Henry**[1,2☯], **Daniel T. Citron**[3☯], **Doreen Mbabazi Ssebuliba**[4], **Juliet Nakakawa Nsumba**[5], **Héctor M. Sánchez C.**[6,7], **Oliver J. Brady**[8,9], **Carlos A. Guerra**[10], **Guillermo A. García**[10], **Austin R. Carter**[1], **Heather M. Ferguson**[11], **Bakare Emmanuel Afolabi**[12,13], **Simon I. Hay**[1,14], **Robert C. Reiner Jr.**[1,14], **Samson Kiware**[15,16], **David L. Smith**[1,14]*

**1** Institute for Health Metrics and Evaluation, University of Washington, Seattle, Washington, United States of America, **2** Quantitative Ecology and Resource Management, University of Washington, Seattle, Washington, United States of America, **3** Department of Population Health, Grossman School of Medicine, New York University, New York, New York, United States of America, **4** Department of Mathematics and Statistics, Kyambogo University, Kampala, Uganda, **5** Department of Mathematics, Makerere University Department of Mathematics, School of Physical Sciences, College of Natural Science, Makerere University, Kampala, Uganda, **6** Division of Epidemiology, School of Public Health, University of California Berkeley, Berkeley, California, United States of America, **7** Division of Biostatistics, School of Public Health, University of California Berkeley, Berkeley, California, United States of America, **8** Centre for Mathematical Modelling of Infectious Diseases, London School of Hygiene & Tropical Medicine, London, United Kingdom, **9** Department of Infectious Disease Epidemiology, Faculty of Epidemiology and Population Health, London School of Hygiene & Tropical Medicine, London, United Kingdom, **10** MCD Global Health, Silver Spring, Maryland, United States of America, **11** Faculty of Biomedical and Life Sciences, University of Glasgow, Glasgow, Scotland, United Kingdom, **12** International Centre for Applied Mathematical Modelling and Data Analytics, Federal University Oye Ekiti, Ekiti State, Nigeria, **13** Department of Mathematics, Federal University Oye Ekiti, Ekiti State, Nigeria, **14** Department of Health Metrics Science, University of Washington, Seattle, Washington, United States of America, **15** Ifakara Health Institute, Dar es Salaam, Tanzania, **16** Pan-African Mosquito Control Association (PAMCA), Nairobi, Kenya

☯ These authors contributed equally to this work.

* smitdave@uw.edu

**Data Availability Statement:** All code is downloadable from: https://dd-harp.github.io/exDE/.

**Funding:** This research was supported by a grant from the National Institute of Allergies and

## Abstract

The Ross-Macdonald model has exerted enormous influence over the study of malaria transmission dynamics and control, but it lacked features to describe parasite dispersal, travel, and other important aspects of heterogeneous transmission. Here, we present a patch-based differential equation modeling framework that extends the Ross-Macdonald model with sufficient skill and complexity to support planning, monitoring and evaluation for *Plasmodium falciparum* malaria control. We designed a generic interface for building structured, spatial models of malaria transmission based on a new algorithm for mosquito blood feeding. We developed new algorithms to simulate adult mosquito demography, dispersal, and egg laying in response to resource availability. The core dynamical components describing mosquito ecology and malaria transmission were decomposed, redesigned and reassembled into a modular framework. Structural elements in the framework—human population strata, patches, and aquatic habitats—interact through a flexible design that facilitates construction of ensembles of models with scalable complexity to support robust analytics for malaria policy and adaptive malaria control. We propose updated definitions for the human biting rate and entomological inoculation rates. We present new formulas to describe parasite dispersal and spatial dynamics under steady state conditions, including the human biting rates, parasite dispersal, the "vectorial capacity matrix," a human

Infectious Diseases (R01 AI163398), which supported DLS, RCR, JMH, and ARC; the grant also funds an active collaboration with the Bioko Island Malaria Elimination Program (BIMEP), which employs CAG and GAG. Additional funding support comes from: a grant from the National Institutes of Allergies and Infectious Diseases (2U19AI089674) that partially supported DLS; a grant from the Bill and Melinda Gates Foundation (INV 030600) that supported DLS, DMS, and JNN; a grant from the National Science Foundation, Directorate for Technology, Innovation, and Partnerships (TIP) as part of the Convergence Accelerator Program (NSF 2040688) that supported DLS and SLW; a UK Medical Research Council Career Development Award (MR/V031112/1) that supported OJB. The research grew out of mosquito working groups discussions as part of RAPIDD (Research and Policy for Infectious Disease Dynamics). Over that time, many of us benefitted from the unwavering support and inspiration of F. Ellis McKenzie. The funders had no role in study design, data collection and analysis, decision to publish, or preparation of the manuscript.

**Competing interests:** The authors declare no competing interests.

transmitting capacity distribution matrix, and threshold conditions. An **R** package that implements the framework, solves the differential equations, and computes spatial metrics for models developed in this framework has been developed. Development of the model and metrics have focused on malaria, but since the framework is modular, the same ideas and software can be applied to other mosquito-borne pathogen systems.

## Author summary

A simple mathematical model of malaria has been the basis for the quantitative study of parasite transmission, but it lacked features to describe spatial dynamics and parasite dispersal. We present a new, modular framework for building highly realistic models of malaria drawing on a century of research and innovation. Using this framework, we develop metrics for parasite dispersal, local reproductive numbers, and malaria connectivity, we re-examine human biting rates and entomological inoculation rates. The framework was built around new, biologically realistic algorithms describing mosquito blood feeding and egg laying in response to resource availability. These algorithms serve as a rigorous yet structurally flexible interface for parasite transmission among human and mosquito host populations; and for the coupled dynamics of volant adult and aquatic immature mosquito populations. The framework supports structured aquatic habitats; patch models for adult mosquitoes; stratified human host populations; and flexible boundary conditions for malaria importation. Using this framework, we can design suites of models with varying levels of realism to study malaria in a place, and we can implement robust simulation-based analytics to support national disease control programmatic activities such as monitoring and evaluation or strategic planning.

## Introduction

*Plasmodium falciparum* transmission dynamics are complex: they involve multiple-agents, non-linear dynamics, localized spatial interactions, spatial, temporal and behavioral heterogeneity, stochasticity, and exogenous forcing by weather, hydrology, and malaria control. Over time, these processes can be modified by economic development; by changing socioeconomic status, human incentives and social norms; and by the evolution of resistance. Every one of these features of malaria transmission dynamics and control presents its own set of challenges to the quantitative study of malaria for scientific research and for analytics to support policy. An important practical problem is how to quantify and synthesize all of the factors affecting transmission at some particular place and time to support malaria control programs in various ways, including monitoring and evaluation of malaria control. The study of complex spatial processes are best addressed using some sort of mathematical model. Here, to fill a need to give robust policy advice, we have developed a modular framework with accompanying software to build and analyze suites of models with scalable complexity for malaria spatial transmission dynamics and control.

A starting point for the quantitative study of malaria transmission dynamics has been the Ross-Macdonald model, which played a central role in developing basic theory and metrics for malaria [1, 2]. That model is simple, general, and conceptually useful, but it is not realistic enough to describe many important features of transmission [3]. The model's lack of realism has also limited its applicability: simple models support generic policy advice, but specific

advice—tailored to context—must be based on models that can quantify and weigh the effects of locally relevant details [4]. A basic limitation of the Ross-Macdonald model was that it lacked features required to describe spatial transmission dynamics and control. Mathematical models for spatial dynamics of mosquito-borne pathogens have been developed [5–18], but there is a need for a generalized synthetic framework to develop and use spatial dynamic models, to extend the Ross-Macdonald model to define and analyze parasite dispersal, to define and measure malaria connectivity [19], and to link spatial dynamics to spatial data. The Ross-Macdonald model is also missing other features that are relevant for malaria dynamics and control, which can be identified from a survey of studies that have modeled mosquito-borne diseases (see Box 1) [2]. Modeling and analyzing real systems can become overwhelming because of computational, parametric, or conceptual challenges that arise from combining all the factors, dimensions, interactions, features, and processes. Individual-based models (IBMs) have been developed around algorithms that make it possible to deal with the complexity by simulating individual states and transitions *in silico* [20], but these high-dimensional computational approaches have some limitations that limit their use and applicability. IBMs require intensive computation, are challenging to parameterize, are difficult to critically evaluate, and their output that is often as difficult to analyze and understand as malaria itself. Using a modular framework, we present an alternative way of dealing with the complexity that is analytically tractable, including some new algorithms to understand mosquito ecology, parasite transmission by mosquitoes, and parasite dispersal on spatial landscapes.

In most places, malaria transmission has been modified by control. The extent of effect modification by malaria control is occasionally revealed when health systems are disrupted (e.g., [21, 22]), when malaria control is relaxed or abandoned [23], or when resistance evolves to drugs or insecticides (e.g., [24, 25]). Programs must weigh evidence and make decisions through analysis of counterfactuals, rather than through direct estimation of control effect sizes, since there would be drastic consequences to experimentally disrupting control. A predominant need in most contexts is thus a set of methods to quantify transmission in its local context as a baseline that has been modified by control. A challenge to achieving this has been that the responses to control efforts are context dependent and have been highly variable across settings. Relevant factors affecting responses to control include details about blood feeding, mosquito ecology, and mosquito behaviors that affect contact with interventions (*e.g.*, resting indoors and IRS). To reconstruct the counterfactual baseline, transmission must be understood in terms of innate mosquito behaviors responding to local resources, vector control, and other contextual factors that have been modified by control. All these have been characterized as being notoriously context dependent and heterogeneous [26–28]. What are the local factors that determine baseline malaria transmission, effect modification, and differences in effect modification at some particular place and time? Basic concerns about the heterogeneous impacts of vector control raise a larger set of questions about how to study and quantify transmission in a way that is relevant for planning malaria control.

This new framework is thus an attempt to bridge two well-established but somewhat contradictory views of malaria. One view is that human malaria transmission dynamics and control are so moulded by local ecology and other conditions that the factors driving transmission or responses to control at one time and place are unlikely to hold elsewhere [27]. Another view —encouraged by the rigorous analysis of the Ross-Macdonald model and extensions of it—is that malaria transmission intensity can be quantified using a small set of bionomic parameters to compute basic reproductive numbers, which also provide a basis for computing threshold conditions for endemic malaria. To build a bridge, the contextual factors affecting basic bionomic parameters must be identified and integrated with new theory describing spatial

**Box 1: Features** This generalized, modular framework presents equations integrating multiple agents and interacting processes. Many of these innovations appeared first elsewhere, but here they are integrated into a single framework:

- Immature mosquito population dynamics structured in distinct aquatic habitats linked to adult populations through egg laying and emergence [31, 32];

- Spatially heterogeneous blood feeding and parasite mixing on vertebrate populations (*i.e.*, blood hosts) with dynamically changing availability, such that feeding rates and the human fraction change adjust to changing conditions [33–36];

- Heterogeneous adult mosquito behaviors, including dispersal, survival, blood feeding, egg laying, mating, and sugar feeding on landscapes in response to spatially heterogeneous resource availability (*e.g.*, mating sites, sugar sources, blood hosts, aquatic habitats) [37–39];

- Multiple vector species or types with different host preferences, daily activity patterns, habitats, *etc.* [40], and potentially with inter-specific resource-based competition in habitats;

- Human mobility based on a concept of time at risk, which combines time spent by humans in places where they are at risk with mosquito blood feeding activity, preferences and other factors [9, 18];

- The capability to model indoor and outdoor spaces for blood feeding, exposure, and vector control;

- A non-linear relationship between the daily entomological inoculation rate (EIR) and the daily force of infection (FoI) due to heterogeneous exposure [41].

- Malaria importation through multiple routes [42];

- An exogenously forced, time-varying extrinsic incubation period (EIP) to model effects of temperature on parasite development;

The model has flexible structural elements to stratify an area into patches, to model any distribution of aquatic habitats, and to stratify a human population into sub-populations by age, immunity, or any heterogeneous, epidemiologically relevant trait. The software also includes time-dependent terms and structures to model exogenous forcing by weather, modification of exposure or transmission by vector control in relation to coverage, including effects of spatial repellents and mosquito behaviors that result in heterogeneous local contact patterns with vector-based interventions.

extensions of the basic metrics, including rigorous, quantitative description of parasite dispersal, and some estimates of the appropriate spatial scales to measure malaria transmission [3].

Context-dependency is an uncomfortable but unavoidable fact of malaria ecology. The heterogeneous nature of transmission and the causes and consequences of variable responses to control have been a difficult and sometimes contentious problem for scientists studying malaria, for national malaria programs and funding agencies making malaria policy, and for

malaria advocates. Historical trends in malaria and the outcomes of malaria control have been so variable that case studies can be found to support rosy projections, alarmist warnings, or contradictory claims about the underlying causes of trends or patterns. To be useful, studies of malaria and programmatic evaluations must acknowledge the important role of context, the multi-factorial nature of causation in these complex systems, non-linear responses to control, the difficulty of measuring heterogeneous systems, and the resulting uncertainty. A consequence of context dependency is the difficulty in drawing conclusions that generalize across systems.

The framework is designed to support development of robust malaria policy advice and to find practical ways of dealing with uncertainty. While scientific research and policy analytics grapple with the same issues and use similar methods, they often put very different weights on uncertainty. Uncertainty affects the ability to do effective inference for scientific research *versus* policy analytics—questions about what is known *versus* what should be done. To address these concerns and give policy advice despite uncertainty, an integrated inferential framework is needed to weigh evidence, integrate the effects of multiple exogenous factors (often involving experts from distinct specialties), estimate their effect sizes, quantify uncertainty, and identify critical gaps. Statistical theory and inferential methods have been developed around the principle of parsimony for scientific inference, but substantially less attention has been given to appropriate designs for analyses that can give advice that is robust to uncertainty. Are the conclusions of an analysis robust to reasonable alternative formulations of a model, and how well are policy recommendations really supported by the evidence? Concerns about robustness could lead to study designs that make different tradeoffs between realism and abstraction. For example, compared with parsimonious models, models with a high degree of realism might be more useful for identifying critical missing data and prioritizing studies to collect it. Robust analytics requires having a modeling framework to build suites of models that are realistic enough to weigh the importance of the major drivers of transmission despite major knowledge gaps.

To address these needs, we have developed a new, modular framework designed to support development of models for robust, simulation-based analytics and adaptive malaria control with scalable complexity. With scalable complexity in model building, members of a model ensemble could range from very simple to very complex, and that models along that spectrum are related to one another through a logical sequence of structural or parametric changes. At one extreme, this framework includes the Ross-Macdonald model, a simple system of differential equations describing the parasite life-cycle in mosquito and vertebrate host populations linked by transmission during blood feeding [1, 29, 30]. By extending the Ross-Macdonald model, simple models can be extended step by step to add complexity or heterogeneity that could be important—based on *a priori* considerations—yet difficult to quantify or poorly informed by existing data (Box 1). With modularity, it is possible to develop new dynamical systems models describing some parts of the system, add or modify components, or add a set of exogenous factors that force a system. It is also relatively straightforward to modify functional responses, or to modify some basic parameters affecting the outcome. Swarms of models can thus be developed to analyze data and to test the robustness of any conclusions. To demonstrate scalable complexity, we here present a complicated family of models that has terms and variables anticipating modification by weather or malaria control. For practical reasons, the model family we present here was scaled back to include a limited set of elements describing transmission, but leaving in place the elements that facilitate modeling control (Box 1). The resulting extensible framework that is capable of describing and analyzing malaria spatial transmission dynamics and control with a high degree of realism in any particular setting. An **R** package which implements the modular differential equations and spatial metrics presented

in the article is available with documentation (https://dd-harp.github.io/exDE/). Despite being programmed in R, the implementation of the mathematical framework into code should be easily adapted to any high-level programming language.

In **Framework**, we first present the modular concepts and structural elements, including a new blood feeding model. Next we present one exemplar model family for each dynamical component. In **Spatial Metrics**, we develop a set of metrics that describe various aspects of parasite spatial dynamics, including metrics for parasite dispersal, connectivity, and the parasite's reproductive success. Finally, in **Quantifying Transmission in a Place**, we discuss the application of these models to the investigation of malaria transmission dynamics and control in a particular place.

## Framework

To describe malaria spatial dynamics with scalable complexity, we designed a modular framework for model building around four core dynamical components, each one a (potentially non-linear) state-space model. An interface rigidly defines interactions among those components, based on passing terms we call *dynamical quantities*. All state variables are vectors of arbitrary length, to accommodate models with different structure or spatial granularity.

To model mosquito ecology, we consider immature mosquitoes in a set of aquatic habitats, and adult mosquitoes in a set of patches. A state space model describes aquatic immature mosquito populations ($\mathcal{L}$) with dynamics $d\mathcal{L}/dt$ requiring an input term from adult mosquito populations: the daily rate eggs are laid in each habitat ($\eta$). A coupled state space model describes mature adult female mosquito populations ($\mathcal{M}$) with dynamics $d\mathcal{M}/dt$ requiring an input term from the aquatic mosquito populations: the rate adults emerge from all the habitats in each patch ($\Lambda$). A state space model for parasite infection dynamics in mosquitoes ($\mathcal{Y}$, which extends $\mathcal{M}$) with dynamics $d\mathcal{Y}/dt$, requires an input term from human malaria epidemiology: the net infectiousness of humans (NI), the probability a mosquito becomes infected after blood feeding on a human (denoted $\kappa$). A state space model describing parasite infection dynamics in humans, immunity, and disease ($\mathcal{X}$) with dynamics $d\mathcal{X}/dt$, requires an input term from adult mosquito infection dynamics: the daily EIR ($E$). The inputs to one component can be passed as trace functions or as the outputs of another coupled component, which is called the *interface* of each dynamical component; a generic interface is coded for each term and if needed specialized methods can be written for particular models. Models in the framework have the following form:

$$
\begin{aligned}
d\mathcal{L}/dt &= F_{\mathcal{L}}(\eta, \mathcal{L}) \\
d\mathcal{M}/dt &= F_{\mathcal{M}}(\Lambda, \mathcal{M}) \\
d\mathcal{Y}/dt &= F_{\mathcal{Y}}(\kappa, \mathcal{M}, \mathcal{Y}) \\
d\mathcal{X}/dt &= F_{\mathcal{X}}(E, \mathcal{X})
\end{aligned}
\tag{1}
$$

The interactions among these dynamical components are thus defined by four input terms ($\eta$, $\Lambda$, $\kappa$, and $E$), which may be computed as outputs of another component or provided as an external forcing term (Fig 1). Because these terms can be computed from the state of the model and are used to couple different model components together, we call these dynamical quantities. These terms are rates which determine how components interact (*e.g.*, flows between components). Because construction of these dynamical quantities can be done in a generic way, computation of these quantities in code can be done for any model which fulfills the interface of its dynamical component.

**Fig 1. Models for malaria transmission dynamics are naturally modular (see Eq 1).** The dynamic modules describe a stratified human population (purple) that interacts through blood feeding (red) with adult mosquito populations in a discrete spatial domain; each patch could contain a set of aquatic habitats. Two components, $\mathcal{L}$ and $\mathcal{M}$, describe mosquito ecology: dynamics of immature mosquitoes (blue) in aquatic habitats are described by a system of equations $d\mathcal{L}/dt$; and dynamics of adult mosquitoes (green) are described by $d\mathcal{M}/dt$. Habitat locations within patches are described by a membership matrix, $\mathcal{N}$. Eggs hatch into larval mosquitoes, that develop, pupate, and later emerge from habitats as mature adults ($\alpha$) and added to the adult populations in each patch ($\Lambda$). Adults lay eggs ($\nu$), which are distributed spatially according to which patch habitats belong ($\mathcal{N}$). Egg deposition rates at the habitats are ($\eta$). Two additional components, $\mathcal{Y}$ and $\mathcal{X}$, describe parasite infection dynamics and transmission: that for mosquitoes, described by $d\mathcal{Y}/dt$ and in humans, described by $d\mathcal{X}/dt$, are linked through parasite transmission. A new model for blood feeding describes how blood meals are allocated among humans ($\beta$) and associated parasite transmission rates: the density of infectious humans by strata ($X$) is used to compute net infectiousness (NI) of humans to mosquitoes in patches ($\kappa$); and the density of infectious blood feeding mosquitoes ($Z$) is used to compute the entomological inoculation rate (EIR) on each strata ($E$).

The dynamical quantities responsible for transfer of pathogens between hosts and vectors are $E$ and $\kappa$, the EIR and NI of humans, respectively. These quantities couple the dynamics between the human $\mathcal{X}$ and mosquito $\mathcal{Y}$ dynamical components. To allow computation of $E$ and $\kappa$ to be highly generic across various types of models of human and mosquito infection, we developed a new model of blood feeding which produces $\beta$, the biting distribution matrix describing how bites arising from mosquitoes at patches are taken on human population strata.

Similarly, the adult $\mathcal{M}$ and aquatic $\mathcal{L}$ mosquito components are coupled via egg laying from adults in aquatic habitats, and emergence of new adults from those aquatic habitats. Because the patches where adult mosquitoes are found may contain many (or no) aquatic habitats, another matrix translates the rate of egg production from adults into egg deposition in each aquatic habitat $\eta$. Likewise, each aquatic habitat produces newly emerging adult mosquitoes at some rate $\alpha$, which in general depends on the current aquatic population, and therefore on lagged adult densities. Another matrix maps this into the rate at which new adults are added to each mosquito population, $\Lambda$.

In addition to reformulating blood feeding and egg laying, the framework includes mathematical descriptions of survival, search for blood hosts or habitats, and dispersal. These new

models of adult mosquito behaviors have all been reformulated around the concept of heterogeneous resource availability and functional responses to available resources.

The modular framework was implemented as a software package, which can be accessed at https://dd-harp.github.io/exDE/, for **R** [43]. The software builds dynamical models of malaria in a modular way using method dispatch to define generic code which implements the framework described here. The dynamical models are functions which return arrays of derivatives of state variables, and can be solved using the integrators available in *deSolve*, or other tools in **R** [43, 44]. The software also includes routines that compute steady state conditions and spatial metrics (see Spatial Metrics, below). Because each component has an interface—the generic functions that compute and pass of dynamical quantities between components—any new model can be implemented which fulfills a specific interface, independent of the rest of the framework. In this way, building and testing new models of particular components is straightforward, and the framework is flexible and extensible. As new models are required, they will be added to the package, increasing its applicability and scope over time.

We have developed a glossary of terms (see S1 Text). In the equations that follow, for each dynamical component, we describe the model structure in detail, and we present one family of models describing transmission dynamics in a single vector species. In a supplement (S2 Text), we formulate a model using both conventional notation and the modular notation of this framework (Box 2). In a vignette to accompany the software (https://dd-harp.github.io/exDE/ ), we have implemented a previously published model of malaria transmission on Bioko Island [45]. In another supplement (S3 Text), we extend the discussion of vector dynamics, including a discussion of models with multiple vector species. All the terms and parameters may be time dependent to accommodate seasonality or modification by exogenous factors: seasonal travel, exogenous forcing by weather, and parameter modification by vector control. Analysis of temporal heterogeneity in this same framework is outside the scope of this study, it but would be straightforward extension following approaches analogous to those shown in the supplements.

## Model structure

The following describes, in detail, the structural elements and the algorithms that connect them. Adult mosquito and human population strata are connected through blood feeding and

**Box 2: Notation** Equations describing spatial processes include terms describing scalar quantities, vectors of scalars, vectors of functions, and matrices. We have avoided using any notation to designate a vector or indicate it could be time-dependent, in part, because it would be ubiquitous; most parameters could vary by space and time. The most general form of a term or parameter is usually described when it is first presented, but most terms describing a vector or matrix should be assumed to be modifiable. In writing out the equations, we consistently use the center dot, "·", in equations to denote the dot product of two matrices, or a matrix and a vector. The juxtaposition of two vectors denotes element-wise product, and 1/* denotes the vector of the inverses of each element. The symbol $\odot$ denotes the Hadamard product (*i.e.*, element-wise multiplication) of two matrices. When $x$ is a vector, $\text{diag}(x)$ is a matrix with the elements of $x$ on the main diagonal. The identity matrix is denoted $I$, and 1 denotes a row or column vector with each element equal to 1. When $F$ is a functional response, we assume it accepts vector arguments and returns a vector of the same length, *i.e.*, $|F(X)| = |X|$. The glossary (S1 Text) discusses the dimensions of each term.

transmission, and adult and aquatic mosquito populations are connected through egg laying and emergence.

**Structural elements.** The framework has been designed to build model ensembles with the goal of studying the spatial transmission dynamics of malaria in a defined geographical area, called the spatial domain. An important part of this framework is having flexibility in defining the model structure to describe spatial and population heterogeneity at the appropriate level of detail, depending on the needs of a study and the available data. The structural elements—the patches, the aquatic habitats, and the population strata—were designed to handle arbitrary patch definitions, arbitrary human population residency patterns and stratification, and arbitrary numbers and locations of aquatic habitats.

To deal with spatial heterogeneity in transmission, we subdivide the spatial domain and identify a set of $p$ patches that includes all locations relevant for studying and quantifying mosquito ecology or transmission: places where people live; places where mosquitoes blood feed; or places with aquatic habitats where mosquitoes lay eggs. We assume that there are $l$ aquatic habitats with actual physical locations that are nested within the patches. To deal with heterogeneity in the human population, the model accommodates stratification. The human population is sub-divided into a set of $n$ population strata by residency, immunity, behaviors affecting risk, or any other epidemiologically relevant factors (S4 Text). Human populations are assigned a single residency patch, where they live and spend most of their nights. Other subdivisions of the human population could take into account age, sex, travel patterns, ITN usage, or any trait that is heterogeneous and epidemiologically relevant. The total census population size, the number of people who reside in each patch in the spatial domain, is given by a vector denoted $P$ (of length $p$). The number of people in each stratum is given by a vector $H$ (of length $n$). In this model, it is not necessary for every patch to have some residents.

To manage terms for interactions among structural elements, we create two mathematical objects called membership matrices that aggregate quantities to patches (S2 Text). Since the $l$ aquatic habitats are scattered among the patches, we define the habitat membership matrix $\mathcal{N}$, a $p \times l$ matrix, that aggregates quantities from the $l$ aquatic habitats to $p$ patches where they are found. Similarly, we define the strata membership matrix $\mathcal{J}$, a $p \times n$ matrix, that aggregates the $n$ human population strata to the $p$ patches where they reside. The census population size, for example, is $P = \mathcal{J} \cdot H$. If a human population were stratified by other traits, such as frequent travel or age, a membership matrix could be created to aggregate model output by trait.

The framework has also been designed to accommodate models with multiple mosquito vector species or types (see S3 Text). Most of the following discussion assumes there is just one vector species, but we point out where the framework has can generalize to multiple vector species.

**Human mobility.** After defining the model structure (*i.e.*, the patches and population strata), the next challenge is to construct the algorithms describing local human mobility and travel. Local mobility determines where and when humans are available and exposed to blood feeding mosquitoes within the spatial domain. We define *travel* in this model by time spent outside the spatial domain; travel and mobility are thus different modalities and handled with different constructs.

To model local human mobility patterns within the patches, we develop a model describing the fraction of time spent by humans in each stratum among the patches [9, 18]. The information is summarized in a time-dependent $p \times n$ matrix $\Theta(t)$, called the *Time Spent* (TiSp) matrix (S4 Text). Each column in a TiSp matrix describes the fraction of time spent in each patch by an individual from a single stratum. In formulating the TiSp matrix, we account for time spent by time of day in the patches where mosquitoes are blood feeding. Total time spent should subtract time spent traveling and and time spent in the spatial domain in places where there is no risk (*e.g.*, in office buildings).

Blood feeding combines human and mosquito behaviors. Since mosquito blood feeding has a daily rhythm [46], time at risk modifies time spent to account for differences in mosquito daily blood feeding activity rates. We let $\xi(t)$ denote a species-specific *circadian weighting function* for blood feeding rates, constrained such that $\int_0^1 \xi(t)dt = 1$, which appropriately assigns a *weight* to time spent by time of day (S4 Text). Using $\xi$, we compute the *Time At Risk* (TaR) matrix as time spent weighted by mosquito activity: $\Psi(t) = \mathrm{diag}(\xi(t)) \cdot \Theta(t)$.

This distinction between TiSp and TaR matrices makes it possible to study human mosquito contact in detail, to quantify differential transmission by multiple vectors with the same human mobility patterns, and to quantify other aspects of mosquito-human contact [47, 48]. A model could have two or more vector species, each with different blood feeding patterns ($\xi_1$ and $\xi_2$), so that one TiSp matrix would be transformed into two different TaR matrices ($\Psi_1 = \xi_1\Theta$ and $\Psi_2 = \xi_2\Theta$).

**Denominators and availability.** After defining host population movement, it is necessary to compute appropriate denominators to model blood feeding, based on the models for time spent and time at risk. Because of mobility, mosquito preferences, and human behaviors, the denominators for blood feeding are different from the resident population size—the number that would be used by most studies (Fig 2).

An important intermediate quantity is ambient population density, which describes the population present in patches at a point in time. In a mobile population, the ambient population density will tend to be different from resident population density. From the time spent matrix, the ambient population density is a vector of length $p$ given by:

$$A(t) = \Theta(t) \cdot H. \tag{2}$$

Similarly, ambient population density at risk is given by: $\Psi(t) \cdot H$. One way to understand what the TiSp matrix means is by taking ratios of ambient to resident populations. The ambient density of residents is $A_r = (\mathcal{J} \odot \Theta) \cdot H$, where $\odot$ denotes the Hadamard (element-wise)

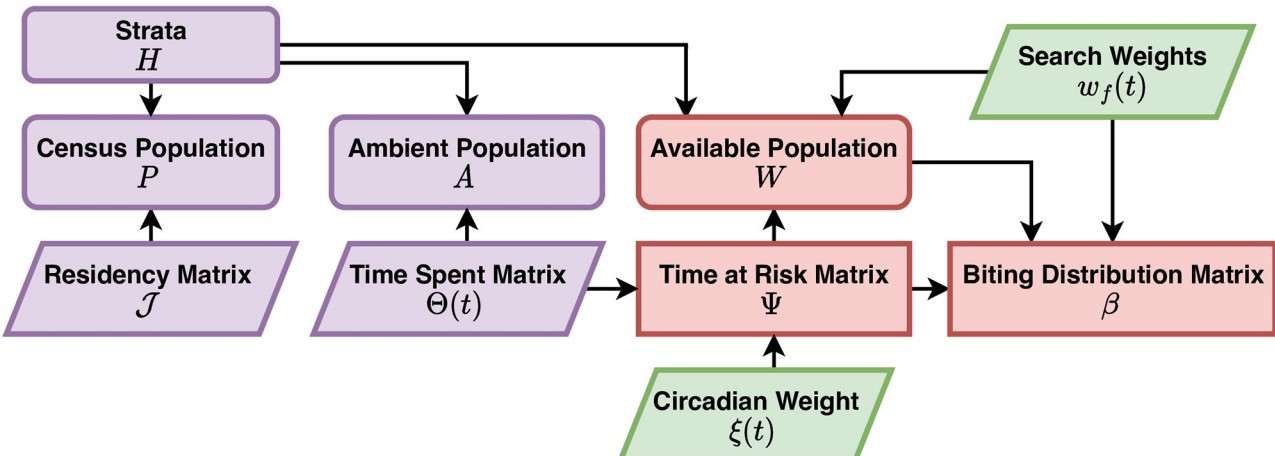

**Fig 2. Denominators and mixing.** A schematic diagram relating various concepts of population density under a model of human mobility, resulting in a biting distribution matrix, $\beta$. Here, and in Figs 3–6, rounded rectangles denote endogenous state variables, sharp rectangles denote endogenous dynamical quantities, and parallelograms represent exogenous data or factors. Purple indicates the element is related to human populations, green for mosquitoes, and red for biting and transmission. Population strata ($H$) describe how persons are allocated across demographic characteristics. The matrix $\mathcal{J}$ distributes these strata across space (patch), according to place of residency. By combining information on how people spend their time across space ($\Theta(t)$) and mosquito activity ($\xi(t)$) a time at risk (TaR) matrix $\Psi$ is generated describing how person-time at risk is distributed across space. Because blood feeding can be modified by human and mosquito factors (*e.g.*, net use and biting preferences), search weights ($w_f(t)$) may further weight person-time at risk. The final result is a biting distribution matrix $\beta$, which is the fraction of each bite in each patch that would arise on an individual in each stratum, so $\mathrm{diag}(H) \cdot \beta = 1$.

product. The non-resident, non-visitor, ambient population is $A − A_r$. The ratios of various census and ambient population densities (*e.g.*, the ratio of residents to ambient population $P/A$, defined wherever $A > 0$), can be used to understand and diagnose unrealistic terms in a TiSp or TaR matrix. The ambient population thus provides one easy statistic to understand TiSp or TaR matrices.

To model the denominators for blood feeding, we also consider other factors—mosquito preferences or human behaviors or traits such as ITN usage—that affect host availability to mosquitoes and relative biting rates on the strata [33]. We assign biting weights, $w_f$, to each strata, where we think of $w_f = 1$ as the value that would be assigned to an average person under baseline conditions (*e.g.*, without a net). These weights affect both the total biting rates and the relative biting rates on the ambient population. We define the *availability* of the host populations to mosquitoes for blood feeding as:

$$W = \Psi \cdot w_f H. \tag{3}$$

Availability is thus defined in units of weighted person-days at risk, and $W$ is a vector of length $p$ describing total human availability in each patch.

We also consider the presence of a population of visitors, a non-resident population spending time in the spatial domain (S4 Text). We assume that some visitors could be present, and that some of them could be infectious. We can let $A_\delta$ denote the ambient density of visitors, but we let $W_\delta$ denote their availability by patch. The *resident fraction* or fraction of human blood meals taken on a resident in each patch, a vector of length $p$ denoted $\upsilon$, is:

$$\upsilon = \frac{W}{W + W_\delta}. \tag{4}$$

The total availability of humans for blood feeding, in each patch, is thus $W + W_\delta$.

**Blood feeding.**  With a well-defined population denominator, we can compute the frequency of blood feeding rates and the human fraction (*i.e.*, the fraction of human blood meals among all blood meals) in each patch in response to the availability of humans and other available vertebrate hosts (Fig 3). To do so, we use functional responses to model blood feeding rates and habits [33–36].

Human availability, $W$, is often highly variable among patches and over time, which could affect the rate mosquitoes blood feed. Mosquitoes could also feed on other vertebrate hosts. To model blood feeding, we supply a vector of functions describing the availability of non-human vertebrate hosts in each patch over time, denoted $O(t)$. We assume that mosquito preferences could scale with host densities, so we assign a shape parameter, $\zeta$, that modifies how preferences scale with host densities. Total availability of all vertebrate hosts for blood feeding is $B = W + W_\delta + O^\zeta$ (S4 Text).

Let $f(t)$ denote the blood feeding rate, the number of blood meals, per mosquito, per day. To guarantee mathematical consistency in computing blood feeding rates (*e.g.*, if $B = 0$, then it should be true that $f = 0$), we can model time-dependent blood feeding rates, where $f(t)$ is a vector of length $p$, as:

$$f(t) = F_f(B) = f_x \frac{s_f B}{1 + s_f B}. \tag{5}$$

Depending on a shape parameter(s), $s_f$, blood feeding rates increase with host availability up to a maximum (or maxima) $f_x$, which is limited by the time it takes to search, process the blood meal, lay eggs, and perhaps to sugar feed. The fraction of blood meals taken on humans at a point in time, a vector of length $p$ denoted $q(t)$, is called the human blood feeding fraction or

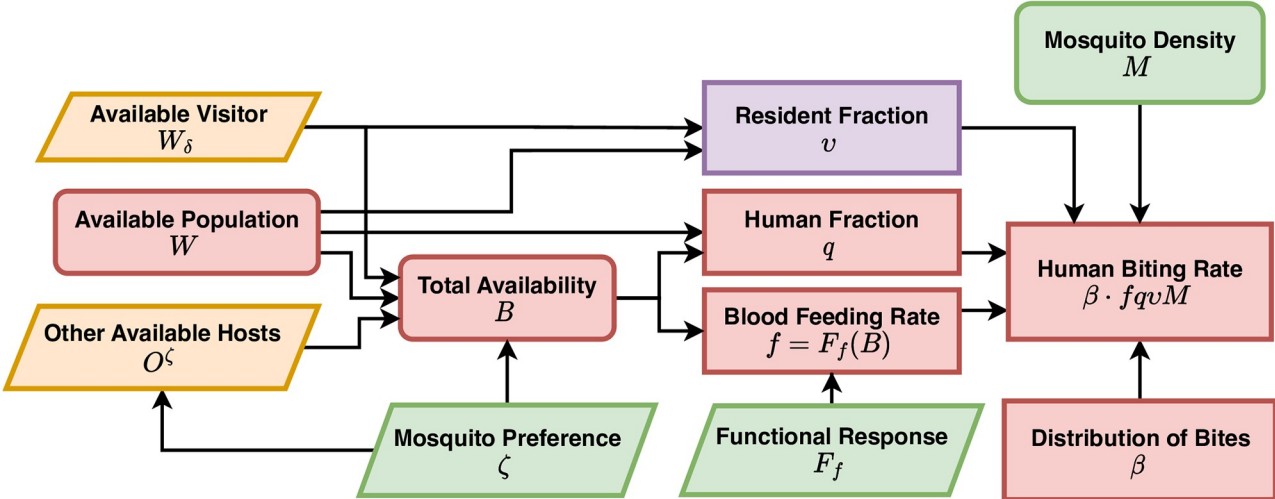

**Fig 3. Blood feeding and human biting rates.** The daily human biting rates (HBR) for the resident population strata are defined as the expected number of bites by vectors, per person, per day. To compute the HBR, we count up exposure over all the patches where residents spend time. We also consider the presence of visitors and other blood hosts (yellow input), which increases the total available hosts.

human fraction:

$$q(t) = \frac{W + W_\delta}{B}. \tag{6}$$

The local human fraction, the fraction feeding on resident humans, is thus $vq = W/B$. The functional forms guarantee that when no humans are present, it must be true that $fq = 0$; and when only humans are available, it must be true that $q = 1$.

**Mixing and parasite transmission.** The model for mixing is an answer to the question: How are blood meals in a patch allocated among humans in the strata? The time at risk matrix and the factors affecting blood feeding rates and habits in each patch must be consistent with the algorithm that computes the distribution of biting and parasite mixing.

To allocate mosquito bites in patches among the resident strata, we let $\beta$ denote a $n \times p$ biting distribution matrix:

$$\beta(t) = \text{diag}(w_f) \cdot \mathbf{\Psi}^T \cdot \text{diag}\left(\frac{1}{W(t)}\right). \tag{7}$$

Each column of $\beta$ describes the fraction of a bite in a patch that lands on an individual in each strata, so the matrix $\text{diag}(H) \cdot \beta$ gives the fraction of bites that land on each stratum, and its columns sum to unity.

In the models for mosquito ecology and infection dynamics, we define variables (vectors of length $p$) for the density of mosquitoes ($M$) and infectious mosquitoes ($Z$). From these, we derive an expression for the daily human biting rate (HBR) and entomological inoculation rate (EIR) for all the strata. The sporozoite rate (SR) in each patch is given by:

$$z = \frac{Z}{M}. \tag{8}$$

The net per-capita human blood feeding rates in each patch, or $fqM/W$, are hereafter called the patch HBR (pHBR), and $fqZ/W$ is hereafter called the patch EIR (pEIR) for infectious mosquitoes. By way of contrast, exposure risk for the strata—the HBR and EIR—are defined as the

number of bites / infectious bites by vectors, per person, per day. The HBR is $\beta \cdot fqvM$, and the EIR is the product of the HBR and the SR, or

$$E = \beta \cdot fqvZ. \tag{9}$$

To draw a sharp contrast between the terms, the pHBR and pEIR describe the number of bites / infectious bites, per person, in patches. They are stratified by location, so they are vectors of length $p$. The HBR and the EIR are stratified quantities that sum exposure over all locations for the strata, so they are vectors of length $n$.

Each model for parasite infection dynamics in humans defines a quantity, $x$, the probability a mosquito becomes infected after biting a human in each stratum. The quantity $X = xH$, a vector of length $n$, is herein called the *infective density* of infectious human residents. We can also specify the probability a mosquito becomes infected after biting a visitor, $x_\delta$. The net infectiousness (NI) for the mosquito populations in all the patches, denoted $\kappa$, is:

$$\kappa = v(\beta^T \cdot X) + (1 - v)x_\delta \tag{10}$$

The force of infection for the mosquito population is thus $fq\kappa$.

**Egg laying.**  To compute quantities affecting mosquito ecology and population dynamics, we need to formulate algorithms to compute egg laying rates and egg laying distributions (Fig 4): how many eggs are laid by adult mosquitoes in a patch, and how are they distributed among the aquatic habitats in that patch? To do so, we develop the concept of habitat availability. We assign a search weight to each aquatic habitat, $w_v$. Using the patch membership matrix, $\mathcal{N}$, we define aquatic habitat availability as:

$$Q(t) = \mathcal{N} \cdot w_v(t) \tag{11}$$

For each patch, total habitat availability is the sum of the search weights for habitats in that patch.

Daily, per-capita oviposition rates of gravid mosquitoes are computed using a functional response to habitat availability, such as:

$$v = F_v(Q) = v_x \frac{s_v Q}{1 + s_v Q}. \tag{12}$$

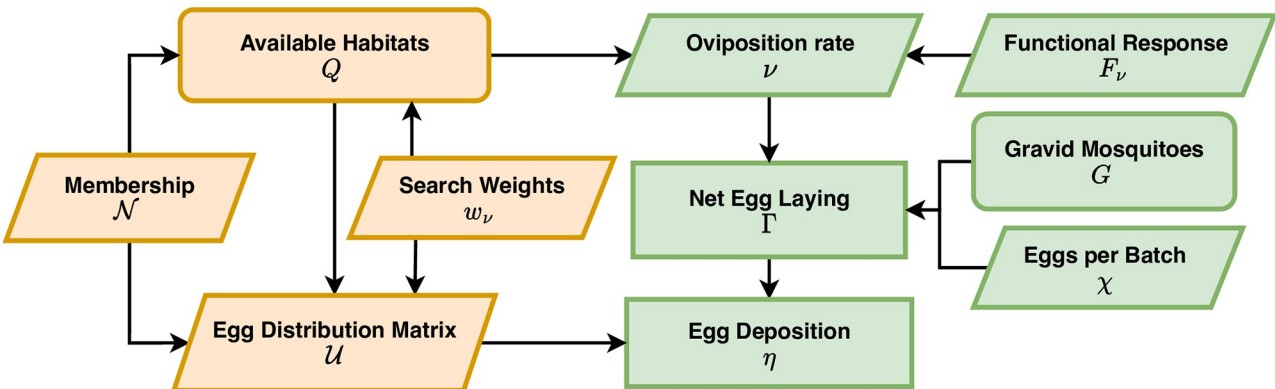

**Fig 4. Egg Laying and egg deposition.** The availability of aquatic habitats ($Q$) the patch sum of habitat search weights ($Q = \mathcal{N} \cdot w_v$), and the egg distribution matrix ($\mathcal{U}$) describes the locally normalized search weights. Available habitat determines per-capita oviposition rates ($v$) by the population of gravid mosquitoes ($G$) in a patch through a functional response to availability, $F_v(Q)$. The net egg laying rate, per-patch, is $\Gamma = \chi v G$. The eggs are distributed among the aquatic habitats ($\mathcal{U}$) so that the egg deposition rates in habitats is $\eta = \mathcal{U} \cdot \Gamma$.

where $v_x$ is the highest possible egg-laying rate for a gravid female, and $s_v$ is a shape parameter. We note that if $Q = 0$, then $v = F_v(0) = 0$. We let $G = F_G(\mathcal{M})$ denote the density of gravid mosquitoes, and we let $\chi$ denote the number of eggs laid, per batch. The net egg laying rate, per patch, per day, is:

$$\Gamma = \chi v G \tag{13}$$

To model egg distribution among habitats, we formulate an egg distribution matrix ($\mathcal{U}$) that allocates eggs to habitats in proportion to local habitat availability. To compute $\mathcal{U}$, for computational reasons we first create $Q^*$ by setting any zero entries to an arbitrary positive value (if $Q = 0$, then $v = 0$, so associated products will later be multiplied by zero), and the egg deposition rate, $\eta$, is computed by:

$$\mathcal{U}(\mathcal{N}, w_v) = \text{diag}(w_v) \cdot \mathcal{N}^T \cdot \text{diag}\left(\frac{1}{Q^*}\right). \tag{14}$$

Finally, we can compute egg deposition rates in the habitats:

$$\eta = \mathcal{U} \cdot \Gamma \tag{15}$$

While $\Gamma$ (a vector of length $p$) describes the net egg-laying rate of the adult mosquito population in each patch, per day $\eta$ (a vector of length $l$) describes the number of eggs laid, in each habitat, per day.

## Core dynamical components

The dynamical quantities whose computation was described above, are configurable elements that connect the four dynamical components: aquatic mosquito ecology; adult mosquito ecology and infection dynamics; and infection and immunity, including human demography. In the following, we describe one model family for each component, including functions that compute terms required for the dynamical quantities; in code these are the generic interface of each dynamical component. These particular models were chosen because they are complex enough to illustrate key features of the framework. These models might not be appropriate for some studies—in particular, the model for epidemiology is too simple for policy. Since the framework is modular, other models can be developed that suit the needs of a study. As part of the software (https://dd-harp.github.io/exDE/), we have formulated alternative model families for some of the components.

**Aquatic mosquito ecology.** The first core dynamical component describes aquatic mosquito population dynamics; the algorithm computes mosquito survival and development from eggs laid through adults emerging. For aquatic population dynamics, we here adapt a previously published model [31, 32].

Let $L(t)$ denote the total density of immature mosquitoes. We let $\psi(t)$ denote maturation rates, $\phi(t)$ the density independent mortality rate, and $\theta(t)L(t)$ describes increased per-capita mortality due to mean crowding. The aquatic dynamics are thus:

$$\frac{dL}{dt} = \eta - (\psi + \phi + \theta L)L \tag{16}$$

The total emergence rate of female mosquitoes in this model, per aquatic habitat, is:

$$\alpha(t) = F_\alpha(L(t)) = \frac{\psi(t)L(t)}{2}. \tag{17}$$

These are recruited into the adult population in the patch, so that the net emergence rate per

patch is:

$$\Lambda(t) = \mathcal{N} \cdot \alpha \qquad (18)$$

While $\alpha$ is a vector of length $l$, $\Lambda$ is a vector of length $p$. This is passed as input to the equations describing adult populations (below).

Given uncertainty about the factors affecting immature mosquito populations, we assume studies might choose to formulate and analyze alternative dynamics. Other dynamical systems models for aquatic ecology in the framework are defined by state variables, $\mathcal{L}$, with dynamics defined by a system of equations $d\mathcal{L}/dt = \eta - F_{\mathcal{L}}(\mathcal{L})$, and a function such that $\alpha = F_{\Lambda}(\mathcal{L})$, such that $\Lambda = \mathcal{N} \cdot \alpha$ (S3 Text).

**Adult mosquito ecology.** The second core dynamical component describes adult mosquito ecology. Given all the functions, terms and parameters above, we have formulated a set of algorithms describing adult mosquito mortality and dispersal that are internally consistent (Fig 5). All this is embodied in the mosquito demographic matrix, called $\Omega(t)$.

We assume mosquito mobility is driven by a search for resources. We have already defined total blood host availability $B$, and aquatic habitat availability $Q$. We also consider sugar availability, $S(t)$, which is passed to the model as a function vector of length $p$. We assume mosquitoes leave a patch while searching for resources, and that they leave a patch more frequently if the resources are less available. Patch-specific emigration rates, $\sigma(t)$, are a functional response to resource availability:

$$\sigma = F_{\sigma}(B, Q, S) = \sigma_x \left( \frac{\sigma_B}{1 + s_B B} + \frac{\sigma_Q}{1 + s_Q Q} + \frac{\sigma_S}{1 + s_S S} \right) \qquad (19)$$

The parameters $\sigma_B$, $\sigma_Q$, and $\sigma_S$ determine the rate that mosquitoes leave a patch if no resources are available, and the shape parameters $s_B$, $s_Q$, and $s_S$ determine how the rate of patch leaving is reduced by the availability of resources. The shape parameter $\sigma_x$ is a scaling parameter that can be used to adjust models with differing patch sizes. Similarly, we formulate a mosquito

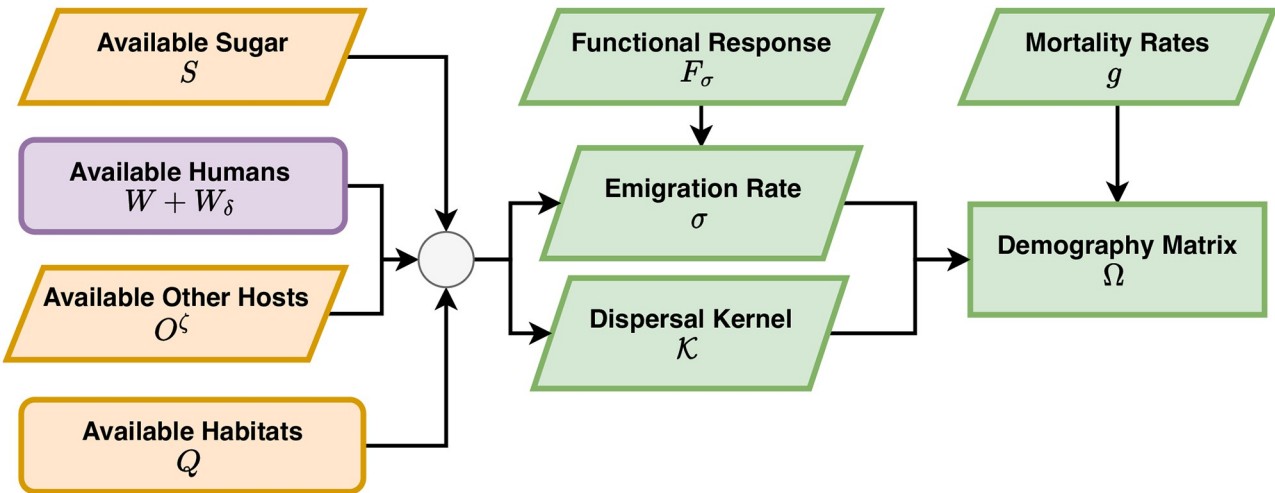

**Fig 5. Adult mosquito demography is defined by survival and dispersal.** Mobility rates and dispersal are determined by the available of resources: aquatic habitats ($Q$), available humans ($W + W_\delta$) and other blood hosts ($O^\zeta$), and sugar ($S$). The emigration rate is a functional response ($F_\sigma$) that increases if any one of the resources is missing. Resource availability and distance also play a role in computing the dispersal kernel, $\mathcal{K}$, that determines where mosquitoes land if they leave a patch. When combined with mortality, a matrix $\Omega$ is produced which describes the behavior of adult mosquitoes after emergence.

dispersal matrix, $\mathcal{K}(t)$ that describes where mosquitoes land after they leave each patch (the diagonal elements of $\mathcal{K}$ are constrained to be equal to zero, S3 Text).

We let $g(t)$ denote the local per-capita mortality rate of mosquitoes in each patch. The matrix $\Omega(t)$ describes adult mosquito survival and dispersal:

$$\Omega = \operatorname{diag}(g) + (I - \mathcal{K}) \cdot \operatorname{diag}(\sigma) \tag{20}$$

where $I$ is the identity matrix.

We let $\Lambda(t)$ be the net emergence rate of mosquitoes into the patches from aquatic habitats (see Eq 18, above). The dynamics of adult mosquitoes are described by the equation:

$$\frac{dM}{dt} = \Lambda - \Omega \cdot M \tag{21}$$

Under the assumptions of this model, the density of gravid mosquitoes, $G$, is:

$$\frac{dG}{dt} = f(M - G) - vG - \Omega \cdot G \tag{22}$$

This model thus assumes that only gravid mosquitoes can lay eggs (Eq 13), but that all mosquitoes (including gravid mosquitoes) can blood feed.

Other models for adult mosquito ecology, denoted $d\mathcal{M}/dt$, could be formulated that describe separate functions for mosquito survival and dispersal, depending on their behavioral states (possibly including sugar feeding, mating and maturation), or that describe a mosquito's reproductive states, or its chronological age or reproductive age. All models developed in this framework must accept the adult emergence rates, $\Lambda$, and they must be formulated in enough detail to compute the population egg-laying rate, $\Gamma$ (see Eq 13).

**Parasite infection dynamics in mosquitoes.** The third core dynamical component describes parasite infection dynamics in adult mosquito populations. Here, we extend a previously published delay differential equation for the density of infectious mosquitoes to include space and a time-varying extrinsic incubation period (EIP) [49].

Let $Y(t)$ denote the density of infected mosquitoes. Using $\kappa$ from Eq 10, the dynamics of infection in mosquitoes are described by:

$$\frac{dY}{dt} = fq\kappa(M - Y) - \Omega \cdot Y \tag{23}$$

We include a time-dependent EIP so that parasite development can be modulated by temperature or other factors exogenous to the system: let $\tau(t)$ denote the EIP for a mosquito that becomes infected at time $t$ (i.e., it becomes infectious at time $t + \tau(t)$ (see S3 Text). We must also define the inverse $\tau^{-1}(t)$, the delay for a mosquito that became infectious at time, $t$. Let $\Upsilon_\tau(t)$ denote a matrix describing survival and dispersal of a cohort from time $t - \tau^{-1}(t)$ through the EIP to become infectious at time $t$:

$$-\ln \Upsilon_\tau(t) = \int_{t-\tau^{-1}(t)}^{t} \Omega(s)ds. \tag{24}$$

When $\Omega$ and $\tau$ are constant, survival and dispersal through the EIP is $\Upsilon_\tau = e^{-\Omega\tau}$. Otherwise, let the $\tau$-subscript denote the value of a variable or parameter at time $t - \tau^{-1}(t)$.

To model the density of infectious mosquitoes, let $Z(t)$ denote the density of infectious mosquitoes. The dynamics of infectious mosquitoes are:

$$\frac{dZ}{dt} = \Upsilon_\tau \cdot f_\tau q_\tau \kappa_\tau (M_\tau - Y_\tau) - \Omega \cdot Z \tag{25}$$

The number of human blood meals per patch, called the net infectious biting rate, is $fqZ$.

Models for infection dynamics, generically denoted $d\mathcal{Y}/dt$ are nested within the model for adult mosquito population dynamics $d\mathcal{M}/dt$ (for example, see [50]). These models accept the net infectiousness ($\kappa$), they must define a variable describing the density of infectious, blood feeding mosquitoes, $Z$, in order to compute the EIR (see Eq 9).

**Epidemiology.** The fourth core dynamical component describes parasite infection dynamics in human populations. Models for malaria infection, immunity, disease, and infectiousness in humans, denoted $d\mathcal{X}/dt$, can become quite complicated, depending on the needs of a study. Studies of malaria epidemiology could consider the complex time course of infections, superinfection, disease, detection, infectiousness, and immunity. The state space describing malaria infection and immunity $\mathcal{X}$ can be modified to suit the needs of a study, and the framework also has enormous flexibility to model heterogeneity in populations through stratification. The following is one model family that is complex enough to illustrate the generic features of the framework.

Let $h = f_h(E)$ denote the local daily force of infection (FoI) and $\delta(t)$ the FoI during travel. In general, $f_h(E)$ could be modified to include heterogeneous biting [41], but in this model, we assume $h = bE$. Both terms are defined for each sub-population. In these models, we stratify on variables relevant for the epidemiology, including immunity, and we model the effects by assigning different parameter values to each stratum.

To model infection dynamics, we modify a hybrid model for the multiplicity of infection (MoI). The dynamics are based on a queuing model, in which new infections occur at the rate $h$, and each parasite clears at the rate $r$, where we track apparent and actual clearance as linked but distinct processes. The variables $m_1$ and $m_2$ track the mean MoI for present and detectable parasites in each strata, which fully describe the epidemiological state space in a simple model with superinfection [51]. We assume that parasites clear at the per-capita rate, $r_1$, so that:

$$\frac{dm_1}{dt} = h + \delta - r_1 m_1 \tag{26}$$

In this model, the true prevalence is:

$$x_1 = 1 - e^{-m_1} \tag{27}$$

We also formulate a model for the MoI of apparent infections. We assume parasite infections are detectable for a shorter time so they appear to clear at a higher rate, $r_2$, and

$$\frac{dm_2}{dt} = h + \delta - r_2 m_2 \tag{28}$$

Similarly, we let $x_2$ denote the apparent prevalence

$$x_2 = 1 - e^{-m_2} \tag{29}$$

We assume that if the infection is patent, a bite infects a mosquito with a higher probability, $c_2$, and $c_1$ if it is not. A bite on a person in each stratum infects a mosquito with probability:

$$x = c_2 x_2 + c_1 (x_1 - x_2) \tag{30}$$

To compute $\kappa$, the infective density of infectious resident hosts by strata is $X = xH$. The vector $X$ is passed to Eq 10 to compute a vector of patch-specific net infectiousness, $\kappa$.

To compute some of the spatial transmission metrics, including basic reproductive numbers (see below), a model must compute the human transmitting capacity (HTC) [52]. In this model, the number of days infecting mosquitoes at the higher probability, $c_2$ is $1/r_2$. The remaining days, are spent infecting mosquitoes at the lower probability. Expressed as the equivalent number of perfectly infectious days, the HTC is:

$$D = \frac{c_2}{r_2} + c_1\left(\frac{1}{r_1} - \frac{1}{r_2}\right) \tag{31}$$

This framework can accommodate other systems of equations describing parasite infection and immune dynamics in humans. This particular model was designed to illustrate some basic features of the modular design. These particular equations were designed to incorporate the effects of immunity on transmission through stratification, allowing parameters describing the duration of infections or detection and the infectiousness to vary among strata (*e.g.*, $r_1$, $r_2$, $c_1$ and $c_2$). New models for human epidemiology can use any epidemiological state space, $\mathcal{X}$, and any system of equations, $d\mathcal{X}/dt$, including models with dynamical changes in the host population size. While the travel FoI is recommended, it is not required. The modules should accept the EIR, and to interact with other components, they must provide a function to compute the infective density of infectious hosts, $X$.

## Spatial metrics

The Ross-Macdonald model defined a set of concepts and metrics that have formed a basis for measuring and understanding malaria transmission, including vectorial capacity and the basic reproductive number $R_0$, but that model and associated metrics did not include metrics for spatial dynamics, parasite dispersal, or malaria importation [3].

Here, we define parasite dispersal by the set of locations (*i.e.* patches) where infecting bites occurred in continuous chains of transmission stretching back in time. Dispersal for any parasite transmission chain is thus defined by locations of the bites that caused each infection, and dispersal alternating between moving humans and mosquitoes between bites. We acknowledge that, due to an observational process, there is an important difference between where an infection occurred and where an infectious person or mosquito is found. There is also an important difference between the formulas defining dispersal and those used to compute reproductive numbers, which count from after a host becomes infectious. Using this definition of parasite dispersal in the context of a model, we have developed formulas and metrics to compute and study parasite dispersal and reproductive success.

To develop these metrics, we assume steady state conditions. This is done for convenience to avoid discussing the complications of understanding spatial dispersal under dynamically changing conditions, and it is a necessary first step to understanding such models. Analysis of malaria transmission dynamics under temporally varying conditions are being developed in a subsequent manuscript.

The formulation of this static model helps to clarify the role of some of the intermediate terms—if all parameters in a model were constant, the transmission model could be fully defined by a much smaller set of parameters, but it may not be clear why the parameters take on those values. Some of the terms that appear in the static analysis correspond to parameters or variables in some Ross-Macdonald models, while others are new: net emergence rates ($\Lambda$) or adult mosquito density ($M$), scaled to the appropriate human population density denominator of host availability ($W$), mosquito bionomics ($f$, $q$, and $\Omega$), and epidemiological parameters

$(r_1, r_2, c_1,$ and $c_2$). New terms describe the spatial biting distribution matrix ($\beta$) and parameters describing malaria importation ($\delta$, $v$, and $x_\delta$).

In models where the context is changing dynamically—due possibly to weather, land use changes, or vector control—exogenous forcing functions can be passed to the model that change resource availability or that perturb the dynamics; the functional forms and intermediate terms (*e.g.* availability) are used to describe changes in the local parameter values and guarantee mathematical consistency. In these static models, the functions and terms are used to set up the model, but after setting parameter values, they need not be called again.

### Net malaria importation and travel fractions

Terms describing the travel FoI ($\delta$) and visitor populations were defined above and integrated into the models for blood feeding and human epidemiology (Fig 6). We define an imported malaria case as a human infection that traces back to a location outside of the spatial domain in the parasite's previous generation, *i.e.*, the mosquito and human host preceding this one in a chain of infections [42]. Net malaria importation rates describe the number of imported malaria cases, per day.

The fraction of all cases that were imported called the *travel fraction* can be defined as either: 1) the fraction of incident infections that were imported; or 2) the fraction of prevalent infections that were imported [45, 53]. To compute these travel fractions, we let $\gamma = (1 - v)x_\delta/\kappa$ denote the *visitor fraction*, the fraction of infectious mosquitoes that were infected by visitors. We let $h$ denote the FoI. The travel fraction for incidence is:

$$\frac{h\gamma + \delta}{h + \delta} \tag{32}$$

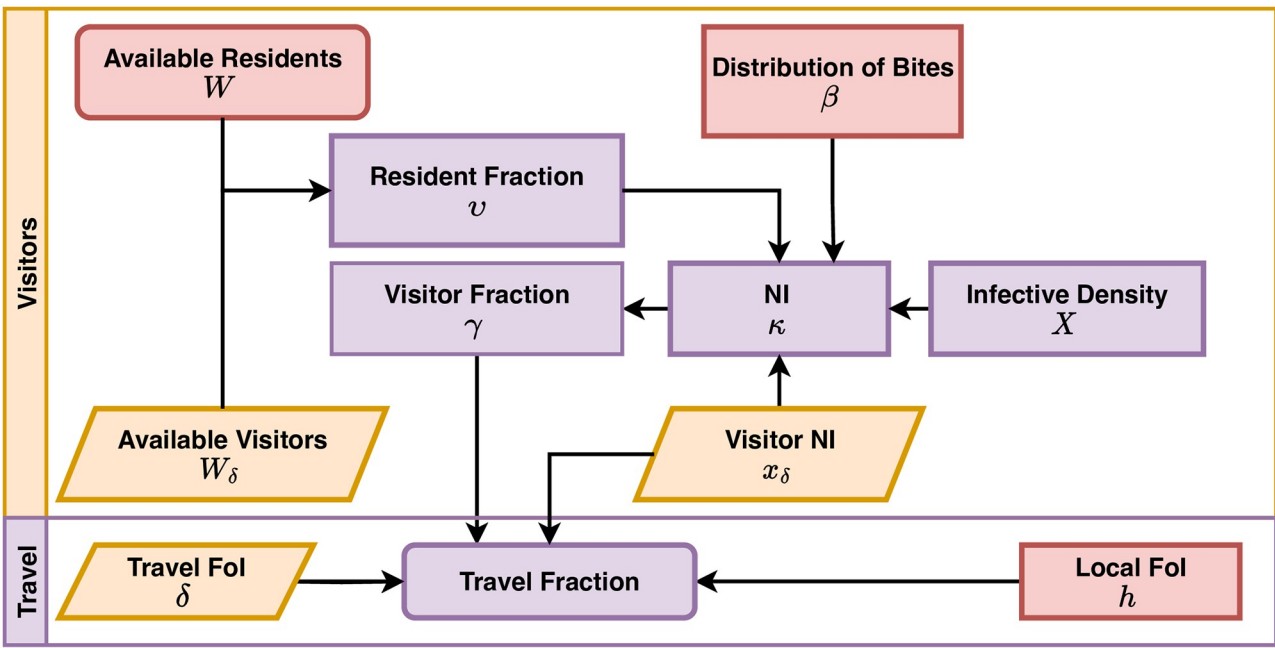

**Fig 6.** To model malaria importation, we define a travel FoI for each stratum, $\delta(t)$, and two set of terms to model the role of visitors in mosquito blood feeding and parasite transmission: the available visitor population $W_\delta$ and the NI for the visitor population, by patch $x_\delta$. To model blood feeding and transmission, we compute a patch-specific resident fraction for blood feeding, $v$, the fraction of all biting that occurs on a resident of the spatial domain. From this, we can compute the visitor reservoir fraction, $\gamma$, the travel fraction for incidence, and other measures of malaria importation.

The travel fraction for true prevalence is:

$$\frac{1 - e^{-(\delta + h\gamma)/r_1}}{1 - e^{-(\delta + h)/r_1}} \tag{33}$$

We note that these are per-capita terms defined for the strata. The net malaria importation rate, the number of imported malaria incidence per day for each patch is:

$$\mathcal{J} \cdot (h\gamma + \delta)H \tag{34}$$

so the travel fraction for incidence for the patches would be:

$$\frac{\mathcal{J} \cdot (h\gamma + \delta)H}{\mathcal{J} \cdot (h + \delta)H} \tag{35}$$

Formulas for the travel fraction for prevalence are formulated in the same way.

## Parasite dispersal

To compute quantities related to parasite dispersal, from bite to bite, we focus on local transmission, and we need some formulas that describe how mosquitoes move around in humans and in mosquitoes.

**Mosquito dispersal and steady states.** In these models, we can compute steady state mosquito population density, assuming $\Lambda$ is constant over time. At the steady state of Eq 21,

$$M = \Omega^{-1} \cdot \Lambda \tag{36}$$

Here, the inverse $\Omega^{-1}$ can be understood as a measure of time spent alive in each patch by mosquitoes emerging habitats in each patch. In other Markov chain models with finite state space, it has also been shown that the elements of the matrix inverse can be interpreted as residence times [54, 55]. In the simpler Ross-Macdonald model, the inverse of a mortality rate, $g$, is a measure of time spent alive or the average mosquito lifespan [56, 57]. The *time spent alive* interpretation of $\Omega^{-1}$ is more apparent if there is no movement: if we set $\sigma = 0$, then $\Omega^{-1} = \text{diag}(1/g)$.

In spatial models, the matrix $\Omega$ accounts for both survival and movement. To illustrate—and to demonstrate that if $\Omega$ is a sensible description of mosquito demography, then the matrix inverse must exist—we construct a tracking matrix. Let $\Xi(t)$ denote a matrix that tracks cohorts of mosquitoes:

$$\Xi(t, M_0) = e^{-\Omega t} \cdot \text{diag}(M_0) \tag{37}$$

It describes the density of mosquitoes left from an initial cohort in each patch $M_0$ that is found in each location at each point in time. There is a duality between the equilibrium population density from Eq 21 and time spent alive by a cohort, computed by integrating Eq 37 (*i.e.* orbits of the related equation $dM/dt = -\Omega \cdot M$). Just as we can compute $g^{-1} = \int_0^\infty e^{-gt} dt$, we can compute:

$$M = \Omega^{-1} \cdot \Lambda = \int_0^\infty e^{-\Omega t} dt \cdot \Lambda \tag{38}$$

so that the steady state can be found by simply adding up the time spent alive in each patch by a cohort emerging from every other patch. Under generalized static conditions (*i.e.* $\sigma > 0$), $\Omega^{-1}$ can thus be interpreted as the average time spent alive in every patch by cohorts of mosquitoes initially found in each patch.

**Parasite dispersal in mosquitoes.** Using mosquito tracking matrices, we can also track parasite dispersal in mosquitoes to derive a matrix that has the same interpretation as the formula for vectorial capacity [57, 58].

To transmit, mosquitoes must blood feed on a human to become infected: the net infection rate in each patch, per available human, is $fq\kappa M/W$. After becoming infected, a mosquito must survive while dispersing through the EIP ($\Upsilon = e^{-\Omega\tau}$). After becoming infectious, a mosquito must blood feed to transmit parasites, so we use the matrix inverse $\Omega^{-1}$ which describes where the mosquitoes are for each infectious human blood meal as long as they remain alive; after becoming infectious, the distribution of infectious bites is given by $fq\Omega^{-1}$. We can describe parasite transmission by mosquitoes by following the story of infection in mosquitoes: after emerging ($\text{diag}(\Lambda)$), a mosquito must blood feed on a human to become infected ($fq\Omega^{-1}/W$); then survive the EIP ($e^{-\Omega\tau}$); and then blood feed to transmit ($fq\Omega^{-1}$).

In the Ross-Macdonald model, the formula for vectorial capacity can be derived from the formula for the daily EIR as a limit [57]. In spatial models, a vectorial capacity matrix can be derived as the limit of a tracking matrix describing the number of infectious bites arising, per available person (*i.e.*, the denominator is $W$), per day at the steady state (S3 Text):

$$\mathcal{V} = \lim_{\kappa \to 0} \frac{fqZ}{W} = fq\Omega^{-1} \cdot e^{-\Omega\tau} \cdot \text{diag}\left(\frac{fqM}{W}\right) \tag{39}$$

Elements in the matrix $\mathcal{V}$ are the expected number of infectious bites eventually arising in every patch from all the mosquitoes in a single patch blood feeding on a single human on a single day, computed as if each human were perfectly infectious. The derivation assumes that no mosquitoes are already infected, and the assumption that humans are perfectly infectious ($\kappa = 1$) is made so that the formula deals only with phenomena related to mosquitoes. In models with multiple vector species, the notion of what it means to be "perfectly infectious" is not as simple because of differences among vector species in their capacity to be a host for the parasites, or vector competence (S3 Text).

**Parasite dispersal by humans.** To quantify parasite dispersal by humans, we compute the human transmitting capacity distribution (HTCD) matrix. We let human transmitting capacity (HTC) describe the net number of perfectly infectious days for each stratum: since infectiousness varies over the time-course of infections, we sum partially infectious days and interpret the HTC as an equivalent number of days spent perfectly infectious [52]. For the population strata in this model, the HTC ($D$) is defined by Eq 31. Since transmission requires two bites, we use the TaR matrix to determine both where a human becomes infected and where it infects a mosquito. Using the transposed TaR matrix, we can describe where infectious days at risk are spent, $\Psi^T \cdot D$. Parasite dispersion by mosquitoes for the sub-populations also accounts for where a mosquito becomes infected, or $b\Psi$.

The HTCD matrix uses the biting distribution matrix, $\beta$, to count from the infectious bite and weight biting appropriately for subsequent blood feeding by all the population strata. The HTCD, a $p \times p$ matrix ($\mathcal{D}$), is:

$$\mathcal{D} = \text{diag}(W) \cdot \beta^T \cdot \text{diag}(bDH) \cdot \beta. \tag{40}$$

We note that $\mathcal{D}$ in spatial models is analogous to $bD$ in models with a single patch. (The equivalency of $\mathcal{D}$ and $bD$ is most apparent if no humans move, and if there is one stratum per patch, and if all search weights are 1, in which case $H = W$ and $\beta = \text{diag}(1/H)$.) Like $bD$, $\mathcal{D}$ describes days spent infectious by an individual human, but in $\mathcal{D}$, describes both where a human got infected and where the mosquitoes were subsequently infected.

The definition of $\mathcal{D}$ as a time-dependent matrix is substantially more complicated if local human mobility patterns change dynamically.

**Parasite dispersal through one parasite generation.** Parasite dispersal is defined by the locations where infecting bites occurred, alternatively moving in infected mosquitoes and

humans. The equations for $\mathcal{D}$ and $\mathcal{V}$ describe the expected movement for a parasite among patches in humans or mosquitoes, respectively, counting from bite to bite. Notably, the formulas are defined for a parasite in either a mosquito or a human. We can also define parasite dispersal through one parasite generation (*i.e.*, from human to human, or from mosquito to mosquito) but the formula depends on where we start counting. If we started from all the mosquitoes blood feeding on a single human (averaged appropriately) on a single day in every patch, then we would get a matrix describing dispersal from every patch to every patch:

$$\mathcal{D} \cdot \mathcal{V}. \tag{41}$$

If we started counting from a typical human infected in a patch on a single day, we would get a different dispersal matrix:

$$\mathcal{V} \cdot \mathcal{D}. \tag{42}$$

Importantly, these formulas follow the same process in the same order, and thus closely resemble the reproductive numbers for malaria (described below), which measure reproductive success for a single parasite. These formulas are two among many that could be developed to count events through a parasite's life-cycle starting at different points.

Formulas that describe the parasite's *per-capita* reproductive success, such as Eqs 41 and 42, counting events arising from a single host. In some cases, we might wish to count the total number of events arising from a patch. To measure the contribution of a patch to overall transmission, we must have a measure of connectivity, or total parasite flows. A tracking matrix describing all of the infections arising from each patch on a day, is:

$$\operatorname{diag}(W) \cdot \mathcal{D} \cdot \mathcal{V} \tag{43}$$

If we started counting infections occurring on humans in a patch, we would get an alternative patch-based tracking matrix. The number of infections arising from a patch is thus tracked by:

$$\operatorname{diag}(W) \cdot \mathcal{V} \cdot \mathcal{D} \tag{44}$$

These measures emphasize the role of places with larger available populations.

The same sort of formulas can be devised to describe transmission from human strata to human strata, but the resulting formulas are only spatial insofar as the human strata are anchored to a residency. If we focused instead on parasite reproductive success starting with an infection in humans, regardless of location, we would get

$$\mathcal{R} = b\beta \cdot \mathcal{V} \cdot \operatorname{diag}(W) \cdot \beta^T \cdot \operatorname{diag}(DH). \tag{45}$$

or we could also count bulk transmission from humans as $\operatorname{diag}(H) \cdot \mathcal{R}$. Notably, Eq 45 is a stratum-based measure. To make it quasi-spatial, we would need to assign events to patches by stratum residency using the membership mapping operator $\mathcal{J} \cdot \mathcal{R} \cdot \mathcal{J}^T$.

**Distances dispersed.** To get a measure of the distribution of distances travelled by parasites, we match a measure of transmission intensity with the corresponding element in a patch distance matrix describing the distance. We take the couplet (distance and intensity) and sort by distance, then compute the cumulative distribution function (CDF). From the CDF, we derive a probability mass function [39]. These dispersal kernels provide a simple way of visualizing distances dispersed by mosquitoes, humans, or parasites.

These formulas and algorithms draw attention to the differences in metrics describing parasite transmission dynamics and dispersal. Because of spatial heterogeneity in mosquito and human population densities, there are many sensible formulas for counting dispersal, some of

which correspond to describing rates, ratios, proportions, and numbers. Careful thought should be given to choosing or developing a metric that fits the analysis.

## Reproductive numbers

Reproductive numbers are a measure of the parasite's average reproductive success. When transmission is spatially heterogeneous, reproductive success will vary for parasites, depending on where they are. As parasites spread over several generations, the expected success of its progeny will change. To calculate threshold criteria for persistence (in the absence of malaria importation), we want a reproductive number to be a measure of average success taken over the whole system, but we want to use an average that does not change across generations. Doing so requires that we compute the *spectral average*, which is computed as the dominant eigenvalue of the parasite's *next generation matrix*.

For many reasons, it is useful to formulate *local* reproductive numbers that describe a parasite's average reproductive success at a particular place and time—an arithmetic average. These *local* reproductive numbers could ignore differences across generations, so they would not serve as thresholds for parasite persistence. In this section, we define local reproductive numbers at the steady state, but the formulas could also serve as point estimates.

Reproductive numbers describe malaria transmission under a range of different conditions that are relevant for understanding malaria transmission dynamics and control or for national strategic planning. Baseline conditions are described by the basic reproductive number, $R_0$, which is defined for a population with no acquired immunity and no malaria control. The adjusted reproductive number, $R_C$, describes a family of numbers defined for a population with no acquired immunity adjusted by malaria control, at a fixed level of control denoted $C$. In other words, $R_0$ is defined as a special case of $R_C$, but in the absence of control. The total effect size of malaria control on transmission is $R_0/R_C$. Here, we also describe the endemic reproductive number, $R_E$, which describes potential transmission modified by immunity. The total effect size of immunity on transmission is $R_C/R_E$. In computing $R_E$, as with $R_0$ and $R_C$, we ignore the fact that some hosts are already infected. In this way, $R_E$ is defined differently than the effective reproductive number, denoted $R_e$, which is lower than $R_E$ because it does not count infections occurring in someone who is already infected. We note that, by definition, at an endemic steady state $R_e = 1$. By way of contrast, $R_E$ counts the number of infections that would occur after one generation, which is useful for planning because it helps to clarify how success in malaria control can be assisted by immunity that will eventually wane.

Both $R_0$ and $R_C$ are computed as if there were no acquired immunity. In this model, the effects of acquired immunity on transmission are quantified through the stratified values of $b$, $r_1$, $r_2$, $c_1$ and $c_2$. These parameters determine the HTC for all the strata ($D$, see Eq 31). If $D$ were computed using values that have been tuned to a stratum with some level of immunity, we would be computing $R_E$. To compute $R_C$, we would need to replace $D$ with values set to a non-immune baseline (*i.e.*, $D_0$), and then recompute the next-generation matrix. Next generation matrices computed with values of $D$ that include the effects of acquired immunity are thus describing an endemic reproductive number. Depending on how $D$ is computed, and whether the bionomic parameters incorporate effects of vector control, we may thus be computing $R_0$, $R_C$ or $R_E$.

**Local reproductive numbers.**   One way to define local reproductive numbers is to modify Macdonald's formula using the local values of parameters, as if there was no movement of mosquitoes or humans. To write the formula using some models in this framework, we may need to modify HTC (which is defined for the strata, of length $n$) to take a patch average. To compute a patch average HTC, $\check{D}$ (a vector of length $p$), we take the population weighted

average,

$$\check{D} = \frac{\Psi^T \cdot w_f DH}{\Psi^T \cdot w_f H} \tag{46}$$

We can then describe a local reproductive number, $\check{R}_C$ (or possibly $\check{R}_E$, depending on how the interpret parameters are defined in $D$):

$$\check{R}_C = \frac{\Lambda}{W} \frac{f^2 q^2}{g^2} e^{-g\tau} \check{D} \tag{47}$$

This local measure is similar to Macdonald's formula [59]. While useful in some contexts, the formula should be applied with caution.

An alternative way to compute local reproductive numbers uses $\mathcal{V}$ and $\mathcal{D}$ (perhaps modified to remove the effects of immunity on transmission). Since the matrices count infections arising from each patch, and we add all infections arising to the patch where the bite originates. We let 1 be a row vector of ones of length $p$, and we can count infections arising starting from all the humans infected in a patch on a single day:

$$\hat{R}_C = 1 \cdot \mathcal{V} \cdot \mathcal{D} \tag{48}$$

that counts infections occurring on humans, or we can start from all the mosquitoes blood feeding on humans on a single day, and:

$$\tilde{R}_C = 1 \cdot \mathcal{D} \cdot \mathcal{V} \tag{49}$$

that counts infected mosquitoes. These patch reproductive numbers could provide valuable information about whether to target the mosquitoes or humans in some patch for enhanced interventions. We could also consider the equivalent formulas for total patch outputs:

$$W^T \cdot \mathcal{V} \cdot \mathcal{D} \text{ or } W^T \cdot \mathcal{D} \cdot \mathcal{V} \tag{50}$$

where $W^T$ is a row vector. Alternatively, we can also weigh transmission from strata using Eq 45:

$$1 \cdot \mathcal{R} \tag{51}$$

or the equivalent scaled by stratum size:

$$H^T \cdot \mathcal{R}, \tag{52}$$

where $H^T$ is a row vector, which gives us valuable information about infections arising from every stratum on every strata, a way of identifying the relative importance of various population strata.

**Next generation matrix.** In the Ross-Macdonald model, a parasite's reproductive success in the next generation is described by a single number. It is computed by counting forward from the moment a mosquito or human becomes infectious. Since parasites move in infected mosquitoes and humans, parasite reproductive success—measured as the number of infections in the next generation—varies across generations as the parasite distributions evolve across generations among strata and among patches. The matrices $\mathcal{V}$ and $\mathcal{D}$ describe parasite transmission and dispersal in mosquitoes and humans, respectively. While the product of these formulas does describe net reproductive success, the computation of threshold conditions has been developed around the concept of a next generation matrix [60, 61], which traces the same process in the same sequence but that start counting at a different point in the parasite's life

cycle (Fig 7). A threshold condition is found by taking the spectral average of the next generation matrix.

In computing next generation matrices, we focus on transmission within a defined spatial domain. For mathematical convenience here, we thus set $v = 1$, though we could easily develop matrices leaving $v$ undetermined to discount exported malaria cases.

We first compute offspring transmitted from a single infectious mosquito to humans or from a single infectious human to mosquitoes, each of which defines a stage in the parasite's next-generation [60]. After a mosquito has become infectious, how many humans (in each stratum) would it infect? In these models, the answer to that question is $n \times p$ matrix, denoted $R_Z$, describing transmission from an infectious mosquito in each patch to humans in each strata:

$$R_Z = b\beta \cdot fq\Omega^{-1}. \tag{53}$$

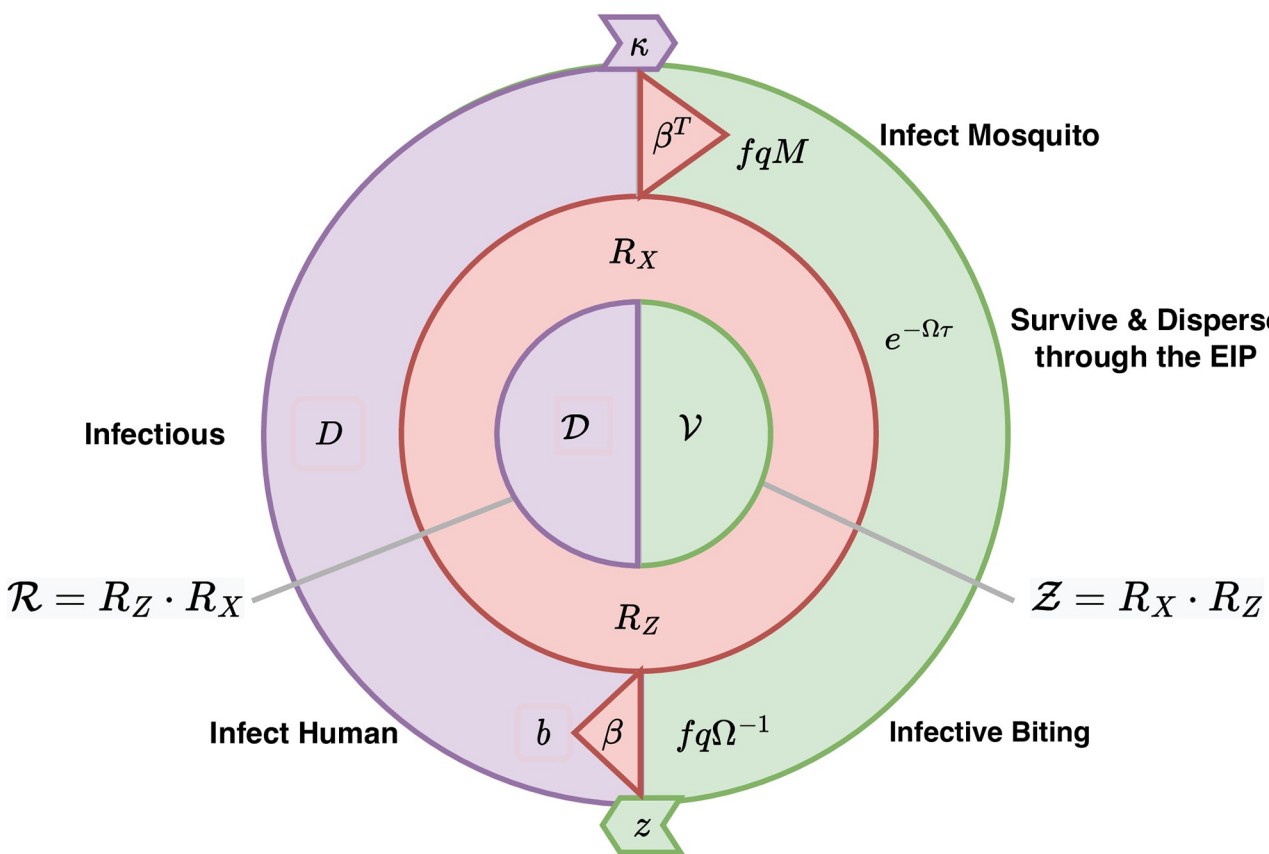

**Fig 7. A spatial life-cycle model.** A diagram that illustrates how the parameters describing each stage in the parasite's life-cycle translate into a parasite's reproductive success spatially, when mosquitoes and hosts move. The right half of the circle represents mosquitoes and the left half humans. The flow of events is clockwise. Mosquitoes must blood feed to become infected ($fqM$), and then survive and disperse through the EIP ($e^{-\Omega\tau}$). infectious bites are distributed as long as a mosquito survives, while it blood feeds and disperses ($fq\Omega^{-1}$). The bites are distributed among humans ($\beta$) and some of them cause an infection ($b$). Parasites are transmitted for as long as humans remain infectious, measured in terms of the human transmitting capacity (HTC, or $D$ days). Infectious humans are distributed wherever humans spend time at risk (affecting $\beta$). These processes are summarized differently to model parasite dispersal and parasite reproductive success. Dispersal counts from bite to bite using the VC matrix ($\mathcal{V}$) and the HTC matrix ($\mathcal{D}$). Threshold computations count from when a host becomes infectious to measure a parasite's reproductive success in infectious mosquitoes ($R_Z$); in infectious humans ($R_X$); from human to humans among strata after a human becomes infectious ($\mathcal{R}$); and from mosquito to mosquitoes ($\mathcal{Z}$). $R_0$ is the lead eigenvalue of $\mathcal{R}$ or $\mathcal{Z}$. Under endemic conditions, we can also consider how frequently parasites are actually transmitted by including the probability a mosquito gets infected $\kappa$, and the probability a mosquito is infectious, given by the sporozoite rate $z$.

How many infectious mosquitoes would arise from each human infection? The answer is a $p \times n$ matrix, denoted $R_X$, describing transmission from a human in each stratum to mosquitoes:

$$R_X = e^{-\Omega\tau} \cdot fqM \cdot (\beta^T \cdot \mathrm{diag}(DH)). \tag{54}$$

The next-generation matrix by type is:

$$\mathcal{G} = \begin{bmatrix} 0 & R_Z \\ R_X & 0 \end{bmatrix} \tag{55}$$

To describe reproductive success in terms of the parasite biology, we count reproductive success through one full parasite generation, either from humans back to humans, or mosquitoes back to mosquitoes. For the parasites, reproductive success through one full generation requires two events, one of each type, so we square the matrix given by Eq 55 to get a new matrix in block form:

$$\mathcal{G}^2 = \begin{bmatrix} \mathcal{R} & 0 \\ 0 & \mathcal{Z} \end{bmatrix}. \tag{56}$$

We thus get two diagonal block sub-matrices describing reproductive success in the parasite's next generation, denoted $\mathcal{R}$ and $\mathcal{Z}$. Reproductive success from human population strata back to human strata is described by an $n \times n$ matrix $\mathcal{R} = R_Z \cdot R_X$:

$$\mathcal{R} = b\beta \cdot \mathcal{V} \cdot \mathrm{diag}(W) \cdot \beta^T \cdot \mathrm{diag}(DH). \tag{57}$$

Reproductive success from mosquito through the population strata back to mosquitoes, described patch-by-patch is described by the $p \times p$ matrix $\mathcal{Z} = R_X \cdot R_Z$:

$$\mathcal{Z} = e^{-\Omega\tau} \cdot \mathrm{diag}\left(\frac{fqM}{W}\right) \cdot \mathcal{D} \cdot fq\Omega^{-1} \tag{58}$$

We have also formulated the next-generation matrix for systems with multiple vector species (S3 Text).

**The spectral average.**   We can also compute $R_C$ as a spectral average through simulation, which is one useful way of illustrating what a spectral average means. To do so, we choose a vector describing the distribution of parasites in a founding generation, $\mathcal{X}_0$ or $\mathcal{Y}_0$, and iterate parasite infections across $i$ successive parasite generations:

$$\mathcal{Y}_{i+1} = \mathcal{Z}\mathcal{Y}_i \quad \text{or} \quad \mathcal{X}_{i+1} = \mathcal{R}\mathcal{X}_i. \tag{59}$$

We define the vector:

$$\mathcal{E}_i = \frac{\mathcal{X}_{i+1}}{\|\mathcal{X}_i\|} \quad \text{or} \quad \mathcal{E}_i = \frac{\mathcal{Y}_{i+1}}{\|\mathcal{Y}_i\|}.$$

where $\|\mathcal{X}\|$ or $\|\mathcal{Y}\|$ is a scalar that denotes is magnitude. Over many generations, $\mathcal{E}_i$ converges to the lead eigenvector, a scalar value also called the spectral average or $R_C$:

$$R_C = \lim_{i \to \infty} \|\mathcal{E}_i\| \tag{60}$$

and it is interpreted as the asymptotic average reproductive success expressed as a number of infected hosts per host, per generation. Note that it is asymptotic only for the linearized system defined by Eqs 55 or 56.

**Quasi-thresholds for endemic Malaria.** Without malaria importation, $R_C > 1$ is a threshold criterion. Analysis of models without malaria importation have consistently demonstrated that malaria is either absent or that there is a single globally, asymptotically stable equilibrium. When there is imported malaria, there are three sufficient criteria for some local parasite transmission to occur within the area:

1. $\max\{\delta\} > 0$ and $R_C > 0$;

2. $\max\{(1 - \nu)X_\delta\} > 0$ and $R_C > 0$;

3. $R_C > 1$.

If condition 1 or condition 2 is satisfied, then malaria will be present in an area, and if $R_C > 0$ then there will be some local transmission. If $R_C > 1$, malaria transmission would be sustained in the absence of importation. We thus call $R_C > 1$ a quasi-threshold for endemic transmission to occur within the spatial domain: *endemic* describes places where $R_C > 1$, and *pseudo-endemic* places where $0 < R_C < 1$ with significant levels of transmission.

## Quantifying transmission in a place

The framework, models developed within it, and the associated spatial metrics were designed to have the skill required to describe and quantify heterogeneous spatial transmission dynamics of malaria in a specific place at a particular time. We have not explicitly defined algorithms for the observational processes that would map model states onto observable quantities, which would be required to extend this mathematical modeling framework into a state space modeling framework to rigorously fit models to data. Instead, we have focused on the mathematics of these processes: time spent by humans; other blood hosts; daily mosquito rhythms; mosquito host preferences, time at risk; and mosquito mobility. Similarly, the models for mosquito ecology and population dynamics describe the mathematics of mosquito mobility, in terms of explicit assumptions about the locations of aquatic habitats, heterogeneous distributions of resources, and mosquito mobility patterns that emerge from a search for resources. By quantifying spatial patterns in terms of the underlying processes—including malaria importation, mosquito ecology and spatial population dynamics, parasite transmission dynamics, human mobility, and malaria epidemiology—the equations point towards a general inferential framework.

Models developed within this framework involve substantially more parameters than the Ross-Macdonald model. This is an inevitable consequence of a decision to model transmission at a particular place and time. If any local features are important for transmission, then a larger set of quantities must be estimated to understand and quantify those features. This gives rise to an important but difficult practical question: *What is the relationship between the amount of local intelligence and the specificity of the policy advice that can be offered?* With minimal local information, it is possible to offer generic policy advice, but it may not be necessary to know everything about a place to tailor advice to context. With this framework, it is possible for models to evolve as the amount information increases, and the models may be used to look ahead to prioritize missing data: *How can programs identify missing information that would most rapidly improve the effectiveness of malaria control?* These contextual factors and the related questions are addressed below.

### Malaria landscapes

While the Ross-Macdonald model describes parasite transmission between abstractly defined mosquito and human populations, the framework we have described was developed to

understand and quantify malaria importation and transmission among structured mosquito and human populations in a well-defined geographical area. (Using a model $d\mathcal{X}/dt$ that describe the infection dynamics of other pathogens and immunity in vertebrate host populations, and making other appropriate choices, the framework could be used as a basis for modeling dengue, West Nile virus, or other mosquito-borne pathogen transmission dynamics, as well.) Since the models are developed to approximate malaria transmission in an actual place, after defining an observational process, the model outputs would be verifiable statements about real quantities over some specific period of time.

As a practical first step, model building starts by defining a set of structural elements—patches, human population strata, and aquatic habits—that are appropriate for the needs of a study (*e.g.* Fig 8 illustrates some options for simulating malaria on Bioko Island, Equatorial Guinea). A geographical study area is usually defined by projects, programs, or political boundaries. In planning interventions for a defined area, an important concern is connectivity to surrounding areas. How much malaria is imported by daily human movement or travel? Are the mosquito populations within the area strongly connected to others nearby?

Using spatial metrics to identify differences in transmission patterns and the flow of parasites across a landscape can help control programs prioritize drugs, outreach, and medical attention to populations, and vector or larval control to places. Using our differential equation framework to reconstruct the equilibrium analysis presented in [45], we have generated spatial bulk transmission matrices (diag($H$) · $\mathcal{R}$) among areas for Bioko Island, Equatorial Guinea. In Fig 9 different patterns of pathogen transport are readily apparent between persons who live in Malabo (left), the densely populated capitol of the island and a sink for travellers, and Luba (right), a small settlement in the Southern half of the island. The pattern of travel seen in Luba typifies most of the areas outside of Malabo, where individuals most often travel to the capitol but not to the other outlying settlements. These patterns affect transmission, where we see parasites originating in Malabo tend to stay in the city. Parasites originating in Luba either tend to stay highly local, or are transported to Malabo when those persons move. Because malarial mosquitoes tend to fare less well in urban settings, these spatial metrics can help understand how high prevalence can be sustained in otherwise unsuitable locations.

An equally important question is about heterogeneity in mosquito population densities within the area and heterogeneity in the risk of exposure, which should inform the definition of patches and the choice of a patch size. Patches, in this model, are defined around adult mosquito activities, and each "patch" has a geographical location. The patch is the spatial unit that defines the algorithms for time spent, blood feeding, egg laying, adult mosquito survival and dispersal. The concept of a patch is flexible enough to model blood feeding indoors and outdoors at the same geographical locations, which may be useful to inform programmatic questions about the effectiveness of vector control measures that target indoor biting (Fig 10). Since the patch is the basis for computing most aspects of blood feeding, the patches define the structure for human time spent at risk, including (if required) quantifying time spent indoors *vs*. outdoors, and mosquito movement rates from indoors to outdoors, from outdoors to indoors, or from outdoors to other outdoor patches.

An important basic concern is the spatial granularity of the patches used for simulation (see Fig 8). Some questions remain unresolved about the appropriate spatial scales and ways to define patches for describing and analyzing malaria transmission for policy (*e.g.*, to compute IRS coverage). One advantage of this framework is that it is possible to build nested models with different spatial grains and compare them. Smaller patches more accurately capture heterogeneity in a landscape while increasing the number of parameters that need to be inferred during calibration to data.

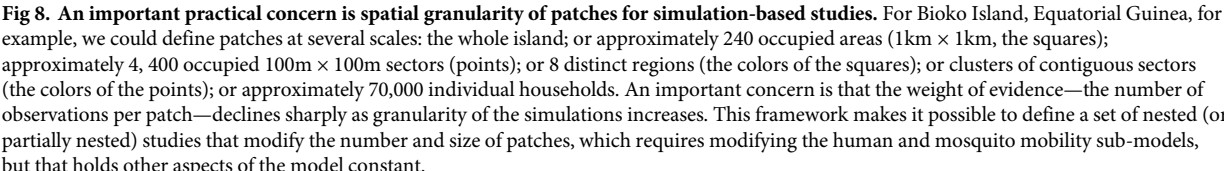

**Fig 8. An important practical concern is spatial granularity of patches for simulation-based studies.** For Bioko Island, Equatorial Guinea, for example, we could define patches at several scales: the whole island; or approximately 240 occupied areas (1km × 1km, the squares); approximately 4, 400 occupied 100m × 100m sectors (points); or 8 distinct regions (the colors of the squares); or clusters of contiguous sectors (the colors of the points); or approximately 70,000 individual households. An important concern is that the weight of evidence—the number of observations per patch—declines sharply as granularity of the simulations increases. This framework makes it possible to define a set of nested (or partially nested) studies that modify the number and size of patches, which requires modifying the human and mosquito mobility sub-models, but that holds other aspects of the model constant.

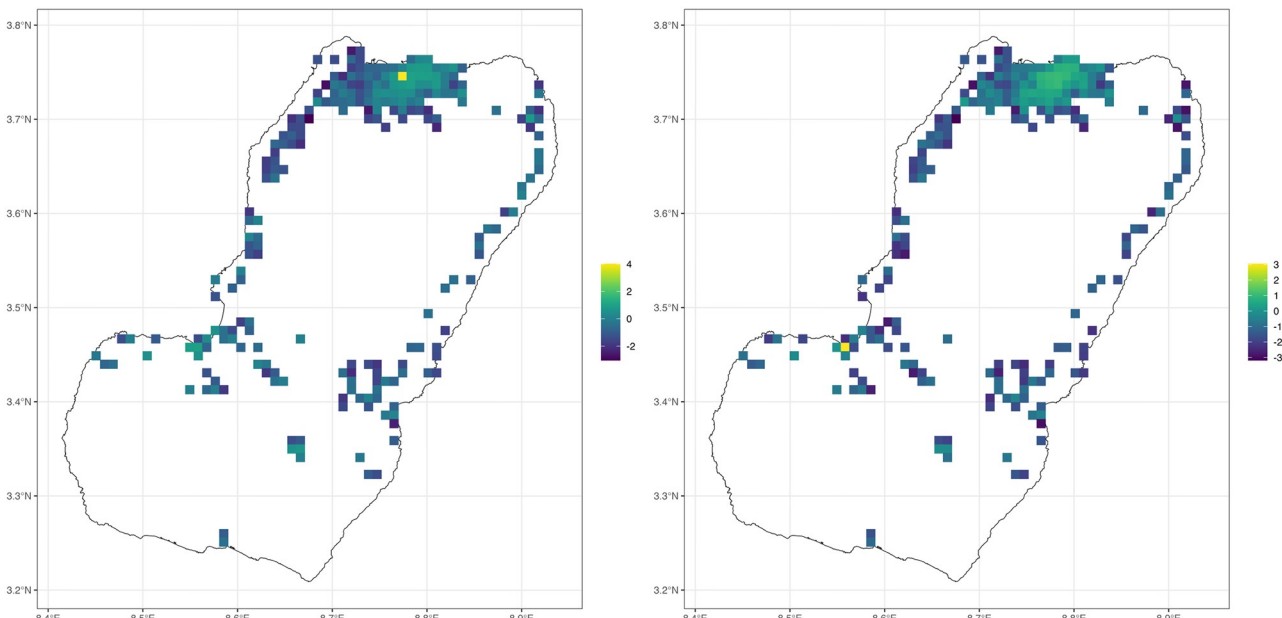

**Fig 9.** (Left): bulk transmission metric describing transmission from the most densely populated area in Malabo, the capitol city, seen as the bright cell in the Northern tip of the island, to all other populated areas. (Right): bulk transmission from the most highly populated area in the south of the island (Luba), seen as the bright cell in the small harbor on the Western coast of the island. The base layer was created by to support malaria control operations [62] and shared under a CC BY 4.0 license. It is [available online] at https://figshare.com/articles/online_resource/Shape_Files_for_Bioko_Island_Equatorial_Guinea/22287580.

Aquatic habitats are located in patches, but the model was designed to assign patches to habitats assuming the habitats had an actual location. Patches in this framework need not have any human residents or any available hosts, so that mosquito habitats in the uninhabited areas around human households are contributing to transmission. Mosquito population dynamics are coupled through related equations describing gravidity, egg laying and egg deposition. The framework thus does not impose any constraints on either the method for constructing patches, or on the number or arrangement of aquatic habitats within the spatial domain. Given the modular nature of these models, the dynamics of immature mosquito populations in each aquatic habitat depend only on its parameters and the egg deposition rates. The productivity of any one aquatic habitat in an area is, however, coupled to other habitats through egg laying by adult mosquitoes that could have emerged anywhere.

To improve the accuracy of models, human populations can be segmented into strata to reduce heterogeneity in traits that affect malaria: the first segmentation is by residency. In this framework, which is designed to quantify process affecting *transmission*, heterogeneity in any trait affecting transmission is dealt with by sub-dividing the population into homogeneous (or less heterogeneous) strata, such as by age, travel habits or patterns, ITN usage, vaccination, care seeking, or any effects of immunity affecting malaria epidemiology or transmission.

Notably, all this structural flexibility is achieved through membership matrices and through the variables describing resource availability, which links search weights, functional responses, and other functional forms to guarantee mathematical consistency (*e.g.* avoiding problems when denominators are zero) despite structural changes. Suites of models can be developed to address concerns about data gaps and uncertainty that are appropriate for studies. Model complexity can be modified by changing dynamical modules, by changing functional forms, by fixing or changing parameters, by splitting and joining patches, by splitting or joining strata, or

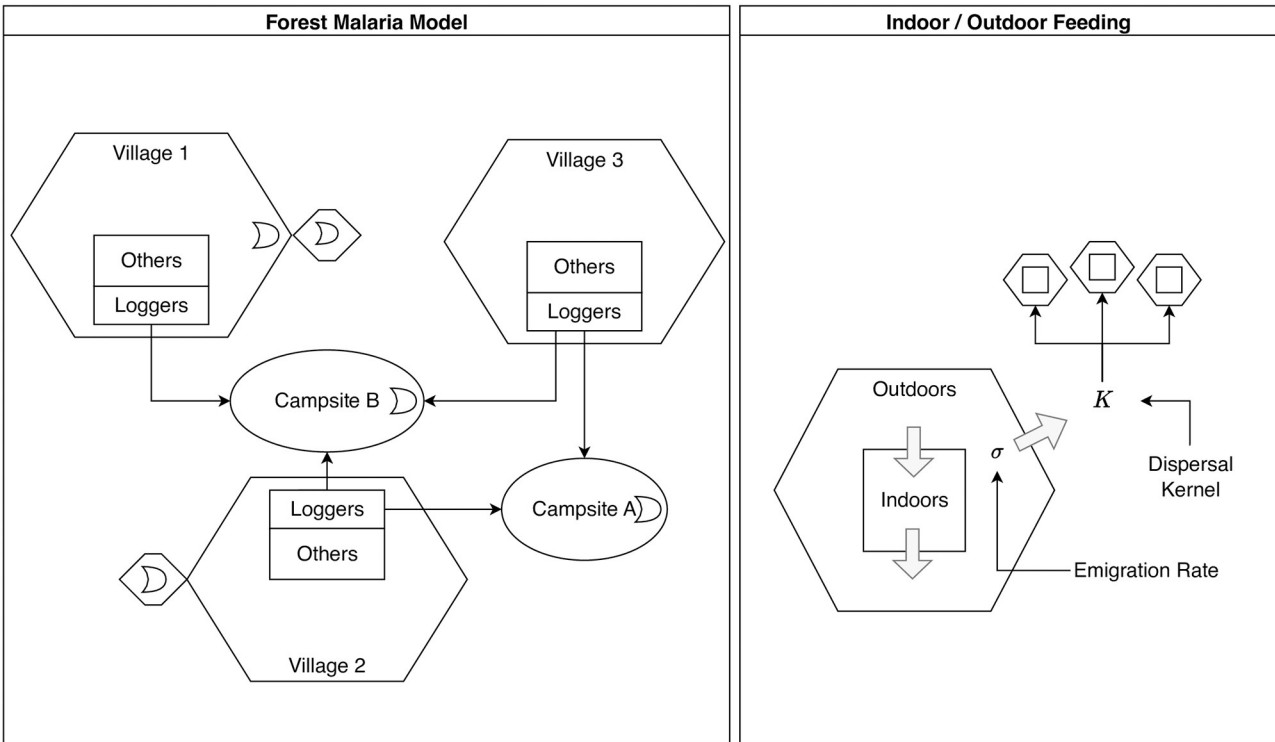

**Fig 10. Structural elements of the framework are flexible to facilitate building models that are appropriate for various settings.** These diagrams illustrate two examples. left) A forest malaria model with seven patches (including 3 villages and 2 campsites), 6 population strata, and 5 aquatic habitats. The village residents are stratified into loggers and other residents. Loggers from different villages spend time at home or in campsites, which have no permanent residents. Aquatic habitats (the moons) can be in villages, in campsites, or in patches near villages. Some villages (*e.g.* village 3), could lack mosquitoes but still have populations at risk. Right) It is also possible to model indoor and outdoor blood feeding with indoor and outdoor patches that share the same place. In these models, movement indoors *vs.* outdoors in the same place is modeled differently from movement among outdoor patches.

by adding and subtracting aquatic habitats. With the ability to split and join patches or strata, any model can be mapped onto simpler, nested models in a series of simple join operations until it is collapsed onto a single-patch, single-stratum Ross-Macdonald model. This is functionally what is meant by scalable complexity.

It is thus as easy to modify and evaluate the effects of model structure (*e.g.* the number of strata) as it is to vary parameters, to facilitate developing suites of models, including models with nested patches or nested strata, to explore tradeoffs in building and calibrating models at various levels of detail.

## Mosquito blood feeding and ecology

Three constant parameters describing mosquito behavior are a standard part of the Ross-Macdonald model [56, 57]: the daily death rate of mosquitoes ($g$), the overall daily blood feeding rate ($f$), and the human blood feeding fraction ($q$). Incorporating the possibility of dynamical feedback between the future emergence of adults and current population size means we have added the population egg-laying rate ($\Gamma$). Adding spatial complexity to the model means the daily emigration rate ($\sigma$), mosquito dispersal ($\mathcal{K}$), distribution of habitats ($\mathcal{N}$) and the distribution of eggs among patches ($\mathcal{U}$) are additional parameters which define how populations may interact in space. While our analysis has focused on steady states, the models were formulated

with parameters that can vary over time in response to changing availability of resources [33–36].

In this framework, the values of all these parameters are computed with functional responses based on resource availability, mosquito biology and innate preferences that constrain the parameters within sensible ranges. This formulation emphasizes how baseline mosquito bionomics for different species could respond to available resources and how those responses would be modified by control. In particular, the same human behaviors can give rise to very different blood feeding patterns for different vector species, depending on the daily rhythms, host preferences, and aquatic ecology of different vector species (S3 Text). We thus have a basis for understanding mosquito behaviors and ecology as a baseline that may have been modified by vector control or weather.

Blood feeding in this model thus makes an important distinction between *anthropophily*, or innate mosquito preferences for hosts of different types, and *anthropophagy*, summarized by the human blood feeding rate (*fq*). Models can also consider a difference between the time of day when mosquitoes are actively searching for blood ($\xi$) and the blood feeding rates by time of day (*f*), which vary with host availability. Innate, species-specific host preferences are embodied in functional forms and parameters, while the rates describing what has happened also depend on context.

Similarly, mosquito population dynamics are an emergent feature of a resource landscape. Since searching for resources is also associated with resource availability, adult mosquitoes will tend to aggregate in patches that have habitats and other required resources. In these models, egg-deposition rates in habitats by volant adult populations are spatially heterogeneous and only partially determined by the emergence rates of adults from a single habitat. The concept of a carrying capacity is, perhaps, not as useful as the concept of habitat productivity and the functional forms that determine how the number of adults emerging is related to the number of eggs laid [31]. A habitat's carrying capacity only makes sense in the abstract—if adult mosquitoes emerging from a single habitat only laid eggs in that natal habitat. In this framework, the aquatic population dynamic module determines how adult mosquito emergence rates respond to egg laying by the adult population.

The parameters describing these processes are both habitat-specific and time-dependent: density-independent mortality, density-dependent mortality, the response to crowding, maturation rates, and search weights could vary for every habitat. A habitat can thus disappear seasonally (which occurs when $w_v = 0$), or weather could affect immature mosquito maturation and mortality rates. If a study called for modeling resource-based competition or stage-structured mosquito populations, the equations describing aquatic populations ($d\mathcal{L}/dt$) can be modified as needed (Fig 1). The framework thus facilitates the construction of realistic models of mosquito ecology, insofar as it is justified by data available and the needs of a study.

## Local exposure, human biting rates and mixing

In defining the algorithms for blood feeding, we also developed a new model for the human biting rate (HBR) and by extension, the entomological inoculation rate (EIR), two basic metrics used to measure malaria transmission entomologically.

The model emphasizes that for any population stratum, the risk of exposure to biting mosquitoes is distributed spatially. In these models, this is determined by a biting distribution matrix ($\beta$). A similar matrix has appeared in other models for the spatial dynamics of mosquito-borne diseases for which human mobility is based on a concept of "visitation" or time spent—classified as Lagrangian movement [7, 8, 10, 12–15, 17, 18, 45]. Here, $\beta$ is based on a concept of availability, the weighted, ambient population at risk. Availability is computed from

observable quantities, and it is computed dynamically for arbitrarily defined human strata and changing availability (the denominator). The formulas guarantee consistency in blood feeding: the number of human blood meals taken by mosquitoes is equal to the number of blood meals received by the humans.

In the new model, the HBR is defined as $\beta \cdot fqM$ and the EIR is $\beta \cdot fqZ$, so that the number of bites received by each stratum depends on how they spend their time at risk. In studies that have reported a value for the HBR or EIR, the quantity reported is based on catch counts by a person or device in a place. In this model, the quantity that is closest to the quantities being estimated is pHBR or pEIR, the number human blood meals, or infectious human blood meals in a patch, per available person, per day ($fqM/W$ or $fqZ/W$). A person who is in a patch at a particular time of day would experience the local biting rates at that time scaled by a search weight ($fqM\xi(t)\omega_f/W$ or $fqZ\xi(t)\omega_f/W$). The quantity being estimated by human landing catches is a measure of the intensity of exposure in a place.

Since other hosts are also available, the number of mosquitoes caught also depends on the biases of the trapping method. In this model, each method for trapping mosquitoes can be thought of as having its own "availability," and it is competing for the attention of mosquitoes. Each method for catching mosquitoes is biased in some unknown way. We thus suggest that field methods designed to estimate the EIR are best interpreted as a location-specific measure of risk in a place, and that epidemiologically relevant measures of risk must acknowledge exposure occurring for a period of time, including all the places where a person spends time. The pEIR, weighted by total availability, is a good approximation of the EIR only if a person spends most of their time at risk in that place. The formulas presented here are useful to quantify how local measures of mosquito blood feeding in a place could differ from what the humans living in that place would experience. What is the difference between risk for a human who moves around compared to their counterfactual self who never leaves home?

## The spatial scales of transmission

Important considerations for planning, monitoring, and evaluating malaria control are the spatial scales that characterize transmission, as defined by parasite dispersal in mosquitoes or humans. We have defined parasite dispersal rigorously in terms of the locations where blood meals occurred that transmitted parasites in dispersal chains. While these definitions are compelling, the distribution of distances separating every pair of infectious bites in a chain of malaria infections can only be approximated using other data. In practice, the framework we have described makes a distinction between local transmission and imported or exported malaria. The framework makes the most sense mathematically if *most* transmission is local, but the framework also defines quantities for malaria importation and exportation, making it possible to study connectivity using a frame that shifts among spatial domains and across spatial scales.

After drawing a bounding box to define a spatial domain and a set of patches, we classify any pair of bites in a transmission chain where at least one occurred in the patch: either both bites occurred somewhere in the spatial domain, called local transmission; or the first bite occurred outside the spatial domain, called imported malaria; or the second bite occurred outside the spatial domain, called exported malaria. These measures of imported and exported malaria thus provide a basis for understanding and quantifying dispersal within and among defined geographical areas. These models weigh the consequences of imported malaria, but as a practical matter, the importance of exported malaria is difficult to quantify because the expected number of subsequent bites depends on conditions somewhere else. Importantly, the fraction that stays local may differ depending on whether the parasite is moving in a mosquito

or a human. Similar definitions and arguments would apply to transmission through a full parasite generation encompassing three bites and two jumps. The metrics we have developed describe transmission within a defined geographical domain, but if there is a need, the models can be reformulated for a larger spatial domain.

The models and metrics provide a way of characterizing the spatial scales of transmission by computing the cumulative fraction of all transmission occurring within a circle of a given radius. Sensible points on that curve can be compared by patch: What distances contain 80%, 90%, 95%, or 98% of all transmission? These estimates are, out of necessity, based on estimated quantities—models of mosquito mobility, human mobility, and modeled mosquito population density—about which there is substantial uncertainty.

Despite the overall uncertainty, these spatial scales are constrained by limits on time and travel. Some quantities are known from census data (*e.g.* population distributions). Most mosquito dispersal distances are short. Mosquitoes can move large distances, but most stay within 1 *km* of a natal habitat [63]. For humans, the fraction of time spent declines sharply with distance away from home. A large fraction of time is spent at home, especially at night, and a larger fraction of the time is spent within roughly 10 *km* of home. The fraction of time spent drops off sharply with $\log_{10}$ distance. The spatial scales also depend on transmission intensity. In places with highly heterogeneous transmission, places with the highest transmission intensity, will have the greater the fraction of transmission that occurs at short distances.

## Mosquitoes, travel, and transmission

Highly spatially resolved data describing the EIR are rarely available. It is often cheaper, albeit less accurate, to use cross-sectional blood survey data describing malaria prevalence (*i.e.* the parasite rate, PR) to estimate local transmission. Spatial models and spatial metrics described herein provide some guidelines about how patterns in the PR can be used to identify areas with the most mosquitoes, particularly given the enormous heterogeneity in human population density.

It is commonly assumed that local clustering of cases implies that there is local transmission. For models developed in this framework, the vectorial capacity matrix (Eq 39) describes parasite dispersion by mosquitoes, and evidence suggests that the spatial scales describing parasite dispersal by mosquitoes could vary by context [63].

Importantly, imported malaria can confound the relationship between local transmission by mosquitoes and prevalence. Travel habits and other traits describing humans often cluster spatially, partly because human neighborhoods are organized by socio-economic status. Spatial clustering of cases could arise if travel habits and thus malaria importation rates are spatially clustered, giving the appearance of local transmission.

## Measuring reproductive success

The most complete measure of transmission in an area is a reproductive number—the number of malaria cases arising from each malaria case after one complete parasite generation. We have defined reproductive matrices in several ways as matrices describing reproductive success among patches within a spatial domain, which can be used to define local reproductive numbers as cases arising from a patch. These reproductive matrices form a basis for investigating the appropriate spatial scales to measure and model transmission, for estimating contamination in randomized control trials, and for understanding the spatial effect sizes of control. These can put other data into a context that is relevant for transmission. For example, mosquito counts data and measures of malaria can vary over very short distances [28, 63]. The functional relevance of local heterogeneity in mosquito catch counts or in malaria prevalence

can be critically examined by examining a matrix that integrates the effects of parasite movement in both mosquitoes and humans. After fully considering the uncertainty, it may be possible to determine the relevant spatial scales of transmission and thus the relevant spatial units for estimating reproductive numbers for malaria dynamics and control.

## Discussion

The simplicity of the Ross-Macdonald model can be contrasted with Hackett's description of the elaborate and context-dependent nature of malaria that he observed in the field [27]:

> . . .*malaria is so moulded and altered by local conditions that it becomes a thousand different diseases and epidemiological puzzles. Like chess, it is played with a few pieces, but is capable of an infinite variety of situations.*

The Ross-Macdonald model clearly identified enough chess pieces to develop basic concepts and theory to describe and measure malaria transmission [1], such as vectorial capacity, the basic reproductive numbers, daily human biting rates, sporozoite rates, entomological inoculation rates, and malaria parasite rates (*i.e.* prevalence). These basic metrics have formed the basis for quantitative studies of malaria transmission, but they ignored heterogeneity and complexity. In particular, the metrics and associated concepts describing *parasite dispersal* in infected mosquitoes and humans were missing.

Parasite dispersal is defined by the locations where infecting bites occurred in chains of transmission, tracing dispersal events backwards through alternating jumps in moving, infected humans and mosquitoes. It is practically impossible to study transmission directly, but this framework has established a quantitative basis for studying transmission through a set of constructs describing closely related processes that can be observed. We have established a basis for describing dispersal rigorously, and for analyzing dispersal and simulating transmission. The metrics and concepts we have proposed here are designed to quantify transmission (and uncertainty about transmission) through the study of patterns and the processes that generated them. The metrics provide a rigorous way of quantifying parasite dispersal and spatial transmission intensity.

In developing models of a specific place for monitoring and evaluating malaria, it is important to understand where and when transmission occurs as well as the local contextual factors that shape transmission. In the Ross-Macdonald model, the basic notions of reproductive success, transmission, and community effect sizes of control were based on the abstract notion of a *population*, but it was never clear how to define a population for purposes of quantifying malaria transmission dynamics: "What, if anything, is a malaria population?" *Focal* transmission has been described [64], but without a quantitative basis for quantifying malaria spatial heterogeneity and spatial dynamics, there was no basis for a nuanced quantitative discussion about "What, if anything, is a focus?" Without defining explicit boundary conditions, it was easy to ignore malaria importation: "What fraction of malaria in a defined area was attributable to local transmission?" Without modeling structured populations, it was impossible to understand how differences in human behaviors would affect transmission [65]. *Who* is responsible for most local transmission or malaria importation? In malaria control, these discussions have focused on the issue of *stratification*, but it remains unclear whether those strata should define sub-populations, spatial areas, or both. Without a framework for understanding malaria transmission spatially in heterogeneous populations, it was difficult to develop a consistent methodology for quantifying transmission in a specific place and time.

We have synthesized a set of old, new, and revised models to fully develop concepts, constrain parameters, and update basic concepts and metrics in a spatial context. New algorithms have filled a need to connect model parameters with data and remove bias while guaranteeing mathematical consistency. The new framework and spatial metrics make model complexity scalable, and it provides a way to study the role of context in mosquito ecology and malaria transmission. How and why do bionomic parameters vary over space and time? What spatial scales characterize mosquito populations? What are the appropriate spatial scales to measure transmission and intervention coverage as a *spatial average*? What are some appropriate methods for dealing with population heterogeneity, including heterogeneity arising from differences in behavior, exposure, or immunity?

The framework emphasizes the way we organize our knowledge about malaria into bins of expertise. Given the complexity of the problem, this means modellers can build models that adapt over time as more information about transmission in a place accumulates. The first models can focus on components whose dynamics are better known, and use simpler, pragmatic approaches to parts of the model whose mechanistic foundations are more uncertain. Our framework can make it easier to build ensembles of plausible models that cover this uncertainty. Model building and model comparison makes it possible to weigh the importance of various factors in context. In asking *where* transmission is occurring, we are concerned about mosquito populations, human behaviors, and human blood feeding. In asking *who* is responsible for malaria, we are not just concerned about differences in infectiousness, but also populations who import malaria, and strata who play an out-sized role in moving malaria around an area. These are the basic quantities that play a role in spatial targeting and in tailoring interventions to context.

## Conclusion

The goal of this study was to develop and present a framework—including mathematical theory and software—to support malaria programs with planning, monitoring and evaluating malaria control. In this manuscript, we describe how to build, solve, and analyze systems of differential equations to model the spatial transmission dynamics of malaria. Suites of models developed in this framework can be used to synthesize data, to quantify the major factors affecting transmission in a particular place, to identify critical data gaps, to prioritize new data collection, to propagate uncertainty through analyses, and to support policy. The spatial metrics and concepts describe an important dimension of malaria transmission that can help tailor and target interventions. In future studies, we plan to address concerns about the temporal dimensions of transmission, including threshold conditions when transmission is seasonal, methods for incorporating forcing by weather and vector control, and the spatial dimensions of malaria control. We plan to use the framework to synthesize evidence and to give robust policy advice about malaria control on Bioko Island, and elsewhere, iteratively as part of adaptive malaria control.

Models developed within this framework—as systems of differential equations—have some advantages and some disadvantages compared to other models. One advantage of this framework is that the models are comparatively easy to understand, modify, and analyze. Because of the modular design, it is possible to build suites of models that start simple and progressively add realism by combining factors from other studies. The framework was designed to lower the costs of building models with arbitrary amounts of realism, so that the model building process is nimble enough to adapt to any problem. We envision this framework as the start of a comprehensive theory for how transmission works, not as a final stage of some trajectory of model development or elaboration. An obvious disadvantage of this framework, however, is

the inability to model stochasticity in the modeled processes, which is especially relevant when population sizes are small, when the disease is invading or near elimination, or when it is important to critically evaluate the measurement of malaria [66]. Individual-based models (IBMs), a commonly used alternative, can handle a great deal of biological complexity and they are implicitly stochastic [67–78]. A disadvantage of IBMs is that the models are harder to understand, that the software constrains the choices—the mechanisms and level of detail—in ways that might not always be apparent to the end user. While the stochasticity matches a feature of the data, there is no guarantee that the IBMs have the right kind of stochasticity, and the noise might obscure other inadequacies of a model. Our theory of transmission generates mathematical constraints between the state variables in the system; while the composed models, constrained by theory, are interpreted here as systems of differential equations, this is not a strict requirement of the framework and future work may explore different mathematical interpretations, including stochastic dynamics.

Policy advice that is based on analysis should go through a rigorous evaluation of its robustness—would the advice change if the analysis had been done in a slightly different but reasonable way? When the advice is based on simulation models, an open question is what kind of model would work best. Ideally, the models would be tested through frequent comparisons to data, but chances to make definitive tests of models against data are rare. Notably, studies of other systems have shown that models with very different underlying mathematics often rank policy options similarly [79]. This is an important kind of study to apply to questions about vector control, disease control, and malaria elimination. Studies should compare models developed within different frameworks and with different levels of detail, through model-model comparison, to identify where the analyses would point to different policy recommendations. A nimble framework to support policy would ideally include the ability to compare deterministic and stochastic models (with various sorts of noise, and demographic stochasticity) with the same level of exogenous forcing by malaria importation, weather, and vector control. An important goal of building frameworks is to conduct studies to identify the appropriate level of complexity through the identification of biologically and policy relevant details [4].

In adaptive management, our goals are to support monitoring and evaluation by developing rigorous methods that quantify malaria transmission as a changing baseline (e.g., forced by weather and other factors) that has been modified by control. In other settings, this framework can be used to enhance the design of randomized control trials or to help programs implement and interpret ad hoc experiments to fill local knowledge gaps. Simulation-based analytics in this framework can be updated using evidence collected by malaria programs to update models and analysis and revise policy recommendations, to target and tailor interventions, and to use evidence to adapt to changing local conditions.

## Supporting information

**S1 Text. Glossary.**
(PDF)

**S2 Text. Modular notation.**
(PDF)

**S3 Text. Vector dynamics.**
(PDF)

**S4 Text. Human travel and mobility.**
(PDF)

## Author Contributions

**Conceptualization:** Sean L. Wu, John M. Henry, Daniel T. Citron, Doreen Mbabazi Ssebuliba, Juliet Nakakawa Nsumba, Héctor M. Sánchez C., Oliver J. Brady, Carlos A. Guerra, Guillermo A. García, Austin R. Carter, Heather M. Ferguson, Bakare Emmanuel Afolabi, Simon I. Hay, Robert C. Reiner, Jr., Samson Kiware, David L. Smith.

**Data curation:** Carlos A. Guerra, Guillermo A. García, Austin R. Carter, David L. Smith.

**Formal analysis:** John M. Henry, Daniel T. Citron, David L. Smith.

**Funding acquisition:** David L. Smith.

**Investigation:** Sean L. Wu, David L. Smith.

**Methodology:** Sean L. Wu, Daniel T. Citron, Doreen Mbabazi Ssebuliba, Juliet Nakakawa Nsumba, Carlos A. Guerra, Guillermo A. García, Austin R. Carter, Robert C. Reiner, Jr., Samson Kiware, David L. Smith.

**Project administration:** David L. Smith.

**Resources:** David L. Smith.

**Software:** Sean L. Wu, David L. Smith.

**Supervision:** David L. Smith.

**Validation:** Sean L. Wu, David L. Smith.

**Visualization:** David L. Smith.

**Writing – original draft:** Sean L. Wu, John M. Henry, Daniel T. Citron, Austin R. Carter, David L. Smith.

**Writing – review & editing:** Sean L. Wu, John M. Henry, Doreen Mbabazi Ssebuliba, Juliet Nakakawa Nsumba, Héctor M. Sánchez C., Oliver J. Brady, Carlos A. Guerra, Guillermo A. García, Austin R. Carter, Heather M. Ferguson, Bakare Emmanuel Afolabi, Simon I. Hay, Robert C. Reiner, Jr., Samson Kiware, David L. Smith.

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
