## [Decision Letter · Decision Letter 0]

7 Feb 2023

Dear Professor Smith,

Thank you very much for submitting your manuscript "Spatial Dynamics of Malaria Transmission" for consideration at PLOS Computational Biology. As with all papers reviewed by the journal, your manuscript was reviewed by members of the editorial board and by several independent reviewers. The reviewers appreciated the attention to an important topic. Based on the reviews, we are likely to accept this manuscript for publication, providing that you modify the manuscript according to the review recommendations.

Sincerely,

James M McCaw, PhD

Academic Editor

PLOS Computational Biology

James O'Dwyer

Section Editor

PLOS Computational Biology

Reviewer's Responses to Questions

**Comments to the Authors:**

Reviewer #1: This paper presents a general mathematical framework for modelling the spatial dynamics of the transmission of falciparum malaria. The paper is incredibly well written and presents the modular framework very clearly. It will be very helpful for anyone new to the malaria modelling field but also for those already in the research area wanting to expand to spatial models. I have only a couple of comments:

• It would be good to have a sense of any other frameworks that are available and how they might compare, e.g. from Institute for Disease Modelling or other groups?

• Since the framework solves systems of ODEs, it would be helpful to have a sense of the contexts that this framework could apply to. Eg control, elimination, both?

Reviewer #2: This is a very well written discription of detailed, yet adaptable / scaleable framework for a spatial malaria model.

The framework is well constructed and appears to be both logically and mathematically consistent and appropriate. I particularly like the very clear and logical separation of structural and dynamic components of the model as well as the explicit spcification of spatial matrices. Another very nice and useful approach the use to the reproductive number to assess the importance of different model components to transmission and their importance for predicting impact of different control measure tragetting individual components.

The challenges with this very nice conceptual framework is that although an implementation in R code, the presentation feel largely a theoretical framework rather than demonstrating its actual implementation. Some of the model component - in particular the spatial structures and the detailed vector components are very detailed and require complex parameterisation (with most parameters being vectors or matrices). Doing this in practice. Defining these parameters will require a lot of data (which may not be available for many other settings) and very extensive model fitting to work in any actual implementation of the model framework. It would be great to see more details on the authors implemented their framework for bioko but I assume that will happen in one or several follow-up publication.

Another minor concern is that in the presented implementation of epidemiologica / within host model is simple. This is sufficient to demonstrate how the differnt components of the framework fit together and interact but - as the authors point out - may have to be substantially more complex for addressing specific public health question.

Lastly, I do have some concerns in regards to who may actually be able to use the R code that is provided. It is very well written, beautiful code. However, it seems to fall a bit in between potential two main user classes. Given that for any extensions/ adaptations specific ODEs need to be specified and programmed, users that are not well familiar with writting and programming ODEs may struggle to adapt the code to their setting and problem. On the other hand highly proficient ODE programmers might find it it (almost) as fast to just program their model from scratch.

However, none of these minor concerns distract from the high quality of the manuscript.

**Have the authors made all data and (if applicable) computational code underlying the findings in their manuscript fully available?**

Reviewer #1: Yes

Reviewer #2: Yes

PLOS authors have the option to publish the peer review history of their article (what does this mean?). If published, this will include your full peer review and any attached files.

Reviewer #1: **Yes: **Jennifer Flegg

Reviewer #2: No

Figure Files:

Data Requirements:

Reproducibility:

References:

---

## [Editor Report · Decision Letter 1]

2 May 2023

Dear Professor Smith,

Thank you very much for resubmitting your manuscript "Spatial Dynamics of Malaria Transmission" for consideration at PLOS Computational Biology.

Having reviewed your response to the reviewers, and noted the minor changes made, before accepting the manuscript I have one final request: Please include in the main text a brief discussion on the second point raised by Reviewer 1 ("Since the framework solves systems of ODEs, it would be helpful to have a sense of the contexts that this framework could apply to. Eg control, elimination, both?"). Your response letter covers the advantages and "obvious disadvantage(s)" (lack of stochasticity so issue in apply to small population sizes and/or when disease is near extinction) but this needs to be clearly and explicitly described in the manuscript itself.

Sincerely,

James M McCaw, PhD

Academic Editor

PLOS Computational Biology

James O'Dwyer

Section Editor

PLOS Computational Biology

Figure Files:

Data Requirements:

Reproducibility:

References:

---

## [Editor Report · Decision Letter 2]

15 May 2023

Dear Professor Smith,

We are pleased to inform you that your manuscript 'Spatial Dynamics of Malaria Transmission' has been provisionally accepted for publication in PLOS Computational Biology.

Best regards,

James M McCaw, PhD

Academic Editor

PLOS Computational Biology

James O'Dwyer

Section Editor

PLOS Computational Biology

---

## [Editor Report · Acceptance letter]

7 Jun 2023

PCOMPBIOL-D-22-01565R2 

Spatial Dynamics of Malaria Transmission

Dear Dr Smith,

I am pleased to inform you that your manuscript has been formally accepted for publication in PLOS Computational Biology. Your manuscript is now with our production department and you will be notified of the publication date in due course.

With kind regards,

Zsofi Zombor
